# Disease-associated gut microbiome and metabolome changes in patients with chronic obstructive pulmonary disease

Kate L. Bowerman [1,5], Saima Firdous Rehman [2,5], Annalicia Vaughan[3], Nancy Lachner[1], Kurtis F. Budden[2], Richard Y. Kim[4], David L. A. Wood[1], Shaan L. Gellatly[2], Shakti D. Shukla [2], Lisa G. Wood[2], Ian A. Yang[3], Peter A. Wark[2,6], Philip Hugenholtz [1,6] & Philip M. Hansbro [2,4,6 ✉]

Chronic obstructive pulmonary disease (COPD) is the third commonest cause of death globally, and manifests as a progressive inflammatory lung disease with no curative treatment. The lung microbiome contributes to COPD progression, but the function of the gut microbiome remains unclear. Here we examine the faecal microbiome and metabolome of COPD patients and healthy controls, finding 146 bacterial species differing between the two groups. Several species, including *Streptococcus sp000187445*, *Streptococcus vestibularis* and multiple members of the family *Lachnospiraceae*, also correlate with reduced lung function. Untargeted metabolomics identifies a COPD signature comprising 46% lipid, 20% xenobiotic and 20% amino acid related metabolites. Furthermore, we describe a disease-associated network connecting *Streptococcus parasanguinis_B* with COPD-associated metabolites, including N-acetylglutamate and its analogue N-carbamoylglutamate. While correlative, our results suggest that the faecal microbiome and metabolome of COPD patients are distinct from those of healthy individuals, and may thus aid in the search for biomarkers for COPD.

[1] Australian Centre for Ecogenomics, School of Chemistry and Molecular Biosciences, The University of Queensland, Brisbane, QLD, Australia. [2] Priority Research Centre for Healthy Lungs, Hunter Medical Research Institute, and The University of Newcastle, Newcastle, NSW, Australia. [3] Thoracic Research Centre, Faculty of Medicine, The University of Queensland, and Department of Thoracic Medicine, The Prince Charles Hospital, Brisbane, QLD, Australia. [4] Centre for Inflammation, Centenary Institute & University of Technology Sydney, School of Life Sciences, Faculty of Science, Sydney, NSW, Australia. [5] These authors contributed equally: Kate L. Bowerman, Saima Firdous Rehman. [6] These authors jointly supervised this work: Peter A. Wark, Philip Hugenholtz, Philip M. Hansbro. ✉email: philip.hansbro@uts.edu.au

Chronic obstructive pulmonary disease (COPD) is a heterogeneous disease with pulmonary pathologies, including chronic bronchitis, airway remodelling and emphysema that impair lung function. It has numerous systemic comorbidities such as cardiovascular disease, colitis and osteoporosis[1,2]. It is the third leading cause of death globally[3], with the primary risk factor being the inhalation of cigarette smoke, air pollution or other noxious particles[4,5]. However, reportedly only 20–25% of smokers develop COPD[6], and while some genetic risk factors have been described[4], other factors such as inflammatory and immune responses are important in pathogenesis[7].

Current approaches to COPD therapy are limited and aim to manage symptoms and reduce exacerbations. High-dose-inhaled corticosteroids are widely employed, but their efficacy is limited to reducing exacerbation frequency or, combined with bronchodilators, improving COPD symptoms[8]. Many patients do not respond to steroid treatment[7], and these therapies fail to modify the factors that initiate and drive disease progression, do not reverse tissue lesions or improve mortality and predispose to serious respiratory infection and pneumonia[8,9].

COPD is punctuated by exacerbations that worsen symptoms. Viruses and bacteria in the respiratory tract are associated with disease exacerbation; however, the heterogeneity of the disease and difficulties in sampling the lung make the exact nature of the relationship difficult to interpret[10,11]. Recently, the respiratory tract microbiome has emerged as a contributing factor in COPD progression outside of exacerbations with substantial overlap in identified viruses and bacteria during stable and exacerbated disease[12]. Comparison of sputum and bronchoalveolar lavage fluid (BALF) between stable COPD patients and healthy controls identified an increased abundance of *Moraxella*, *Streptococcus*, *Veillonella*, *Eubacterium* and *Prevotella* in disease[13,14]. However, other studies of BALF reported increased *Prevotella enoeca* but no difference in *Streptococcus*[15]. Comparisons of lung explants identified increased *Proteobacteria* and reduced *Firmicutes* and *Bacteroidetes* with decreased abundance of *Streptococcus*, *Haemophilus influenza* and *Prevotella* spp. in COPD[16]. Reduced bacterial diversity occurs in stable COPD patient sputum compared to healthy controls[13]; however, both increased and consistent diversity has been reported in BALF[14,15]. These studies suggest that the lung microbiome does not reproducibly change in COPD, which may be related to its transient nature produced by the balancing forces of immigration and elimination that typically impede long-term colonisation[17,18].

The co-morbidity of colitis suggests that the 'gut-lung axis' may be important in COPD pathogenesis[19]. Thus, we hypothesised that changes in the permanently colonised gut environment may contribute to pathogenesis and be a more reliable indicator of COPD. The concept of the gut-lung axis, describing the common mucosal immune system of the lung and gastrointestinal tract, implicates roles for the gut microbiome in regulating inflammation in acute and chronic respiratory disease including COPD[18,19]. Several studies implicate disturbances in the abundance or metabolism of gut bacteria in asthma and allergic airway disease[20–22]. In addition, the gut microbiome regulates host immune responses to respiratory infection[19,23], and may, therefore, contribute to exacerbation frequency in COPD. COPD patients have increased incidence of gastrointestinal disturbances such as ulcerative colitis and Crohn's disease and vice versa[24,25], indicating potential roles for the gut microbiome in the disease. However, the gastrointestinal microbiome of COPD patients has not been assessed[26,27].

Here we compare the composition and functional potential of the gut microbiome in COPD patients with those of healthy controls, using untargeted faecal metagenomics and metabolomics. We describe an altered gut microbiome and metabolome associated with the disease. Several strepotococci and members of the family *Lachnospiraceae* discriminate between COPD patients and healthy controls in addition to correlating with impaired lung function. The metabolomic analysis identifies a shortlist of metabolites that may be potential biomarkers for validation in future studies. These findings support the gut microbiome and metabolome as being altered in association with COPD and highlight the need for further exploration of this environment to uncover whether it plays an active role in disease progression via the gut-lung axis.

## Results

**Participant profiles**. We separately characterised the gut microbiome and metabolic profiles in COPD by analysing stool from individuals satisfying the global initiative for chronic obstructive lung disease (GOLD) criteria and healthy controls. A total of 28 COPD patients (54% female) and 29 healthy controls (66% female) were assessed, all during periods of stable disease (Supplementary Data 1). Information on GOLD status, dietary habits, smoking status and medication history was collected, along with spirometry and blood cell counts (Supplementary Data 1–5). COPD patients include four classified as GOLD I, 11 as GOLD II, eight as GOLD III and 11 as GOLD IV. The COPD cohort was older than healthy controls (mean age of 67 vs. 60, $p = 0.012$) and had a significantly higher proportion of past smokers ($p = 0.005$). Daily fibre intake was lower in COPD patients, while pulse rate, total white blood cell, neutrophil, monocyte and eosinophil counts were significantly higher. No significant differences were observed in body mass index (BMI), systolic or diastolic blood pressure, the proportion of current smokers, daily energy, carbohydrate, fat, protein, sugar or starch intake, haemoglobin, total red blood cell, platelet, lymphocyte or basophil counts between the groups (Supplementary Data 1).

**Faecal microbiome taxonomic indicators of COPD using 16S rRNA gene sequencing**. To compare the gut bacterial community composition between COPD patients and healthy individuals, we initially undertook 16S rRNA gene sequencing. In total, 4285 sequence variants were identified across all 57 faecal samples. After filtering for sequence variants present in at least two samples with a minimum relative abundance of 0.05%, 977 sequence variants were retained for community analysis. A significant difference in overall community composition was observed between COPD and healthy gut microbiomes (Fig. 1a, $p < 0.0001$ PERMANOVA of Bray–Curtis distances), without a significantly altered level of diversity ($p_{Shannon} = 0.329$, $p_{SimpsonInverse} = 0.291$). COPD status explained 4% of the between-sample variability indicating substantial inter-individual differences that remained largely uncaptured by the addition of further demographic variables (Supplementary Data 6). Sequence variants contributing to the distinction between the groups were identified using multivariate sparse partial least-squares discriminant analysis (sPLS-DA, Fig. 1b–d, Supplementary Data 7). Genera increased in abundance in COPD include *Streptococcus* and *Rothia*, both common oral bacteria as well as occurring in the gut[28], *Romboutsia* and *Intestinibacter* from the family *Peptostreptococcaceae* and *Escherichia*. Genera decreased in COPD include *Bacteroides*, *Roseburia* and *Lachnospira* from the family *Lachnospiraceae* and several unnamed genera of *Ruminococcaceae*.

**Faecal microbiome taxonomic indicators of COPD using metagenomics**. Having identified distinct COPD-associated faecal taxa using 16S rRNA gene sequencing, we sought to increase the resolution of these findings via metagenomic sequencing of the same samples. We recovered 437 metagenome-assembled

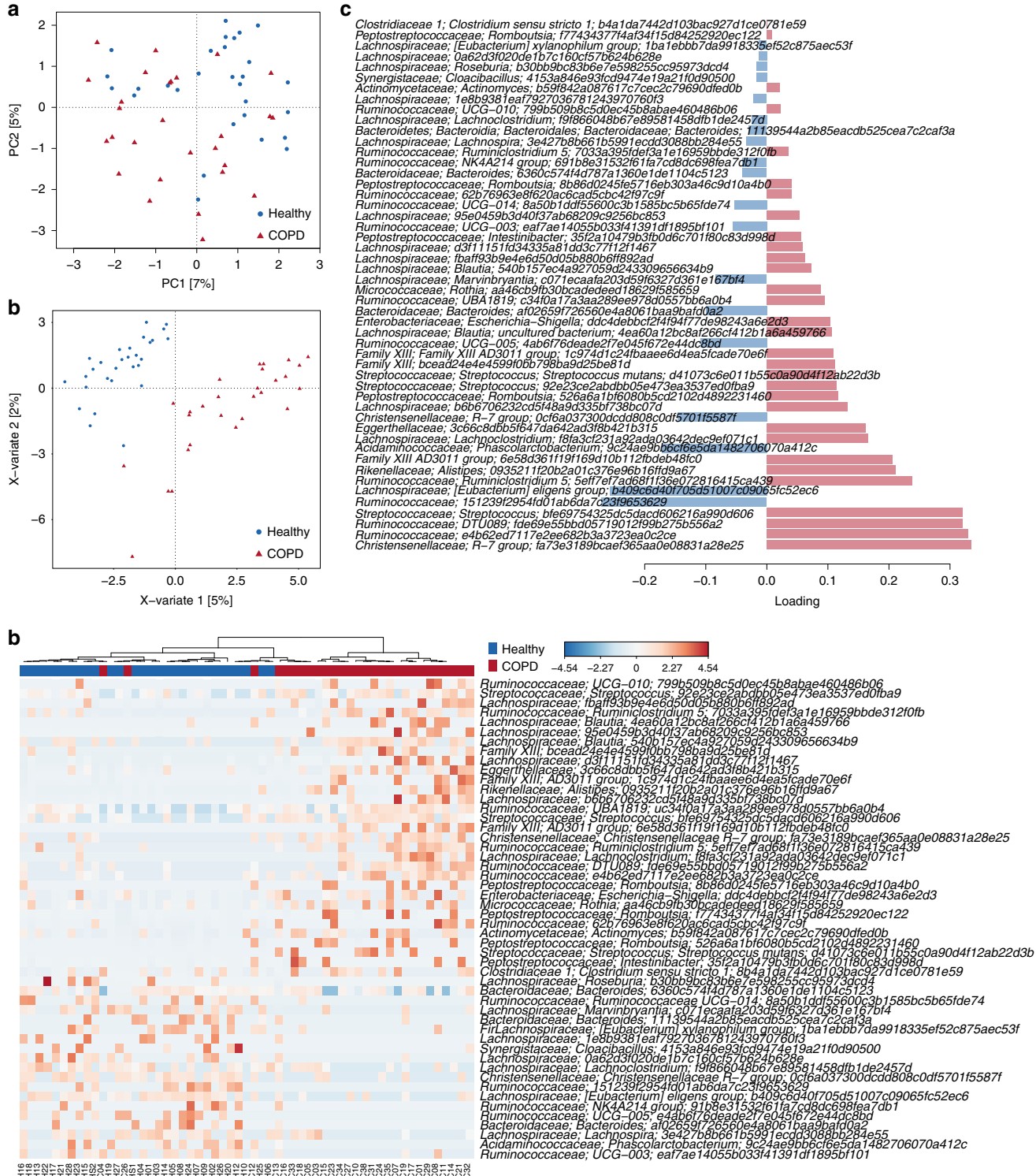

**Fig. 1 Faecal microbiota of COPD patients ($n = 28$) can be distinguished from that of healthy individuals ($n = 29$) using 16S rRNA gene amplicon sequencing. a** Principal component (PC) analysis undertaken at the sequence variant level using read counts transformed using log-cumulative-sum scaling. **b** Multivariate sparse partial least-squares discriminant analysis (sPLS-DA) of read counts transformed using log-cumulative-sum scaling at the sequence variant level. **c** Sequence variants contributing to separation along with component 1 of sPLS-DA from **b**. Bar length indicates loading coefficient weight of selected genomes, ranked by importance, bottom to top; bar colour indicates the group in which the sequence variant has the highest median abundance, red = COPD, blue = healthy. **d** Heatmap of read counts transformed using log-cumulative-sum scaling of discriminatory sequence variants identified along with component 1 of sPLS-DA from (**b**).

genomes (MAGs) from 57 individuals, each with an estimated completeness >80% and a maximum of 7% contamination. Overall community composition was analysed using these MAGs in combination with a set of publicly available reference genomes. Consistent with the 16S rRNA gene sequencing analysis, COPD and healthy samples could be distinguished (Supplementary Fig. 1a, $p < 0.0001$, PERMANOVA of Bray–Curtis distances) despite considerable variation in community composition between individuals (Supplementary Fig. 1b) and no significant differences in diversity between the groups ($p_{Shannon} = 0.174$, $p_{SimpsonInverse} = 0.345$). COPD status explained 6% of the between-sample variability (Supplementary Data 8). At the bacterial family level, *Bifidobacteriaceae*, *Eubacteriaceae*, *Lactobacillaceae*, *Micrococcaceae*, *Streptococcaceae* and *Veillonellaceae* were enriched in COPD. Depleted families included *Desulfovibrionaceae*, *Gastranaerophilaceae* and *Selenomonadaceae* along with several uncharacterised families of *Bacilli* and *Clostridia* (Supplementary Data 9). Enriched and depleted families were highly variable between individuals (Supplementary Data 9 and Supplementary Fig. 1b), as is frequently observed with human datasets[29].

To identify genera and species contributing to the distinction between COPD and healthy controls, we employed both univariate and multivariate approaches designed to identify significantly different species (DESeq2[30]) and the largest source of variation between the two groups (mixOmics[31]), respectively (Fig. 2a–c). Over 200 genomes belonging to 107 genera and 146 species were identified as either significantly enriched or depleted between COPD and healthy samples using DESeq2 although the differences in average relative abundance for most species were small (Supplementary Data 10). Some species were present at a substantially higher prevalence in COPD patients including *Rothia* and *Streptococcus* spp., *Romboutsia timonensis* and *Intestinibacter bartlettii*, consistent with 16S rRNA gene sequencing, while others were more prevalent in healthy controls (e.g. *Coprobacter fastidiosus* and *Coprobacter secundus*, *Rikenellaceae* genus *RC9* and *Christensenellales* family *CAG-74*). *Streptococcus* species were identified as key differentiators between COPD and healthy samples using sPLS-DA analysis within mixOmics, as were multiple members of the family *Lachnospiraceae* (Fig. 2b, c).

**Microbiome changes indicate disease status**. To test whether patient characteristics contributed to the microbiome signature separating COPD from healthy controls, we repeated the univariate analysis of the metagenomic data, including age, BMI and sex within a multifactorial design in DESeq2, categorising BMI according to WHO standards and age in 10-year windows (≤54, 55–64, 65–74 and ≥75). *Streptococcus vestibularis*, and two unnamed *Streptococcus* species (*sp001556435*, *sp000187445*) remained significantly enriched in COPD samples using this model, and *RC9* genomes remained enriched in healthy samples (Supplementary Data 11). We compared medication-related subgroups within the COPD samples and found no significant difference in microbiome composition between those taking inhaled steroids, beta-agonists or anticholinergics and those not taking these drugs ($p = 0.286$, $0.208$ and $0.220$, respectively, PERMANOVA of Bray–Curtis distances). There was also no significant difference between current smoking and non-smoking COPD patients ($p = 0.224$, PERMANOVA of Bray–Curtis distances) or between stable and frequent exacerbators ($p = 0.367$, PERMANOVA of Bray–Curtis distances). Correlation analysis revealed a subset of taxa that were significantly associated with lung function. These included negative correlations between *Streptococcus sp000187445* and *S. vestibularis* and forced

expiratory volume in 1 s (FEV$_1$) and most COPD-associated members of the family *Lachnospiraceae* with predicted per cent-forced vital capacity (FVC) and FEV$_1$ (Fig. 3). Positive correlations were observed between *Desulfovibrio piger_A* and *CAG-302 sp001916775* and lung function. Overall, these data support an association between the faecal microbiome and COPD status, identifying species associated with both health and disease; there are some associations with disease severity, as indicated by blood neutrophils, lung function and historical frequency of exacerbation episodes.

**Functional potential indicators of the COPD faecal microbiome**. Metagenomic reads were annotated with predicted function based on alignment against available databases (Pfam, TIGRFAM, KEGG and CAZy), for a gene-centric analysis of unassembled metagenomes. There was no significant difference in overall predicted functional capacity between COPD and healthy samples in a global comparison of all annotated domains (Supplementary Fig. 2). However, pairwise comparison at the individual domain level revealed several annotated functions that were distinct between the two groups. Glucosyltransferase enzymes were enriched in COPD based on enrichment of domains in each database: PF02324 (Pfam), TIGR04035 (TIGRFAM), K00689 (KEGG) and GH70 (CAZy) (Supplementary Data 12–15). These enzymes synthesise high-molecular-weight extracellular glucan polymers such as α-D-glucans from sucrose that adsorb onto the bacterial surface and contribute to the adherence of *Streptococcus* and other species[32]. LPXTG-anchored adhesion domains (K12472 and TIGR04225), a cell-surface-anchoring motif found in Gram-positive bacteria, were also enriched in COPD samples. Most of the reads annotated as containing the enriched domains aligned to the enriched *Streptococcus* populations (Supplementary Data 16–19). Glucosyltransferase-annotated reads aligned to *S. salivarius* and *Streptococcus sp001556435 gtfC* genes, of which there are multiple copies within the enriched reference genomes. LPXTG-anchored adhesion domains were identified within a YSIRK-type signal peptide-containing protein in *S. salivarius*, *S. parasanguinis_B* and other *Streptococcus* spp. (Supplementary Data 20 and 21). The protein also carries multiple CshA-type fibril repeats used by *Streptococcus gordonii* to bind fibronectin[33]. Fibronectin is expressed by epithelial cells and is upregulated in murine models of colitis and in association with inflammatory bowel disease[34,35]. Increased fibronectin is observed in the small airways of COPD patients[36] and in experimental COPD[37]; however, no similar analysis is available for the gut. The capacity for adhesion to host tissue may therefore contribute to the enrichment of streptococci in the COPD gut microbiome.

We then also undertook a targeted genome-centric analysis comparing the encoded functions within genomes identified as significantly different between COPD and healthy samples in either multivariate or covariate-adjusted univariate analyses (35 enriched in, and 25 depleted in COPD relative to healthy controls, Fig. 2b and Supplementary Data 11). The majority of the predicted discriminatory functions were encoded in genomes enriched in COPD (Supplementary Data 22). These included *Streptococcus*-specific features such as the accessory secretory proteins Asp1–3, forming part of the accessory SecA2/Y2 secretion system that exports glycosylated serine-rich repeat glycoproteins involved in adhesion[38]. Also specific to *Streptococcus* are the typical streptococcal peptidoglycan biosynthesis enzymes (penicillin-binding proteins, murN) and an ABC-type manganese uptake system involved in streptococcal virulence[39]. Elements of multiple amino acid biosynthesis pathways were also enriched among COPD-associated genomes, as were fatty acid biosynthesis initiation and elongation enzymes. Genomes

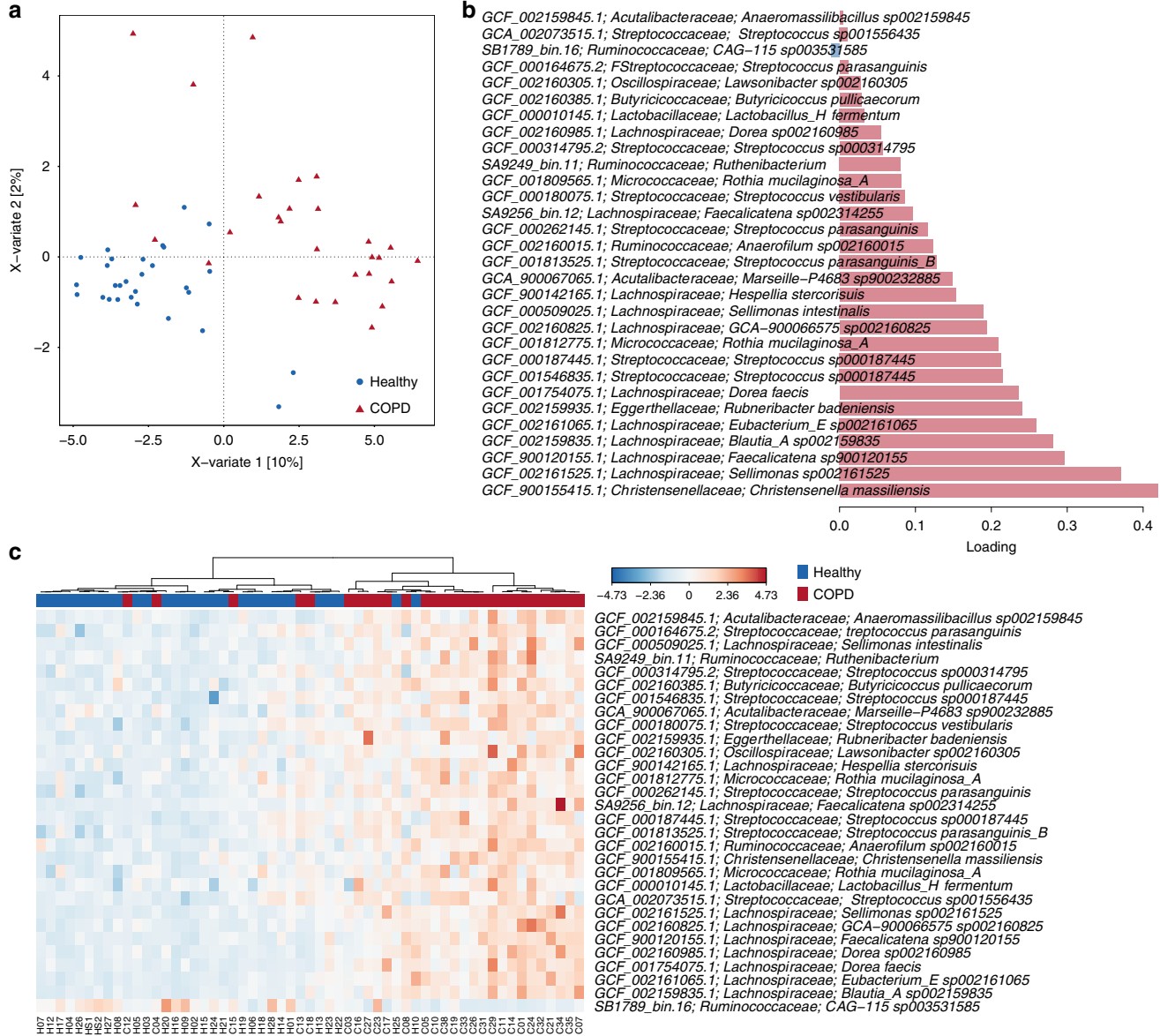

**Fig. 2 Metagenomic sequencing-based exploration of COPD-associated ($n = 28$) faecal microbiomes supports distinction from those of healthy individuals ($n = 29$). a** Multivariate sparse partial least-squares discriminant analysis (sPLS-DA) of read-mapping-based relative abundance at the genome level of the faecal microbiome, filtered for genomes with minimum 0.05% relative abundance in at least one sample. **b** Genomes contributing to separation along with component 1 of sPLS-DA from (**a**). Bar length indicates loading coefficient weight of selected genomes, ranked by importance, bottom to top; bar colour indicates the group in which the genome has the highest median abundance, red = COPD, blue = healthy. **c** Heatmap of discriminatory genomes along component 1 of sPLS-DA from (**a**). Data are centred with log-ratio-transformed relative abundance.

associated with healthy samples from the uncharacterised families *CAG-138* (order *Christensenellales*), *CAG-239* (order *RF32*), *CAG-1000* (order *RF39*), *CAG-302* (order *RF39*) and *CAG-508* (order *TANB77*) lack many of these functions based on KEGG module completeness (Supplementary Data 23), as recently observed amongst uncultivated members of the gut microbiome[40]. They may therefore represent gut symbionts reliant on host metabolites making them potentially more sensitive to environmental perturbation.

**Functional indicators of the COPD faecal metabolome.** To assess metabolic expression in the COPD gut, we undertook untargeted metabolomic profiling of paired faecal samples identifying 934 compounds likely arising from both the microbiome and the host, and some from ingested compounds

(Supplementary Data 24). Principal component analysis (PCA, Supplementary Fig. 3a) revealed significant but incomplete separation of COPD and healthy samples ($p = 0.003$, PERMANOVA of Euclidean distances). As with the metagenome, there was no significant difference in the metabolome of COPD patients between those taking steroids, beta-agonists or anticholinergics and those not ($p = 0.299$, 0.724 and 0.596, respectively, PERMANOVA of Euclidean distances), between current smokers and non-smokers ($p = 0.115$), or stable and frequent exacerbators ($p = 0.501$).

**Integration of metagenomes and metabolomes.** We used the mixOmics platform to both investigate the metabolites contributing to the distinction between COPD and healthy samples and to integrate the metagenomic and metabolomic data into a

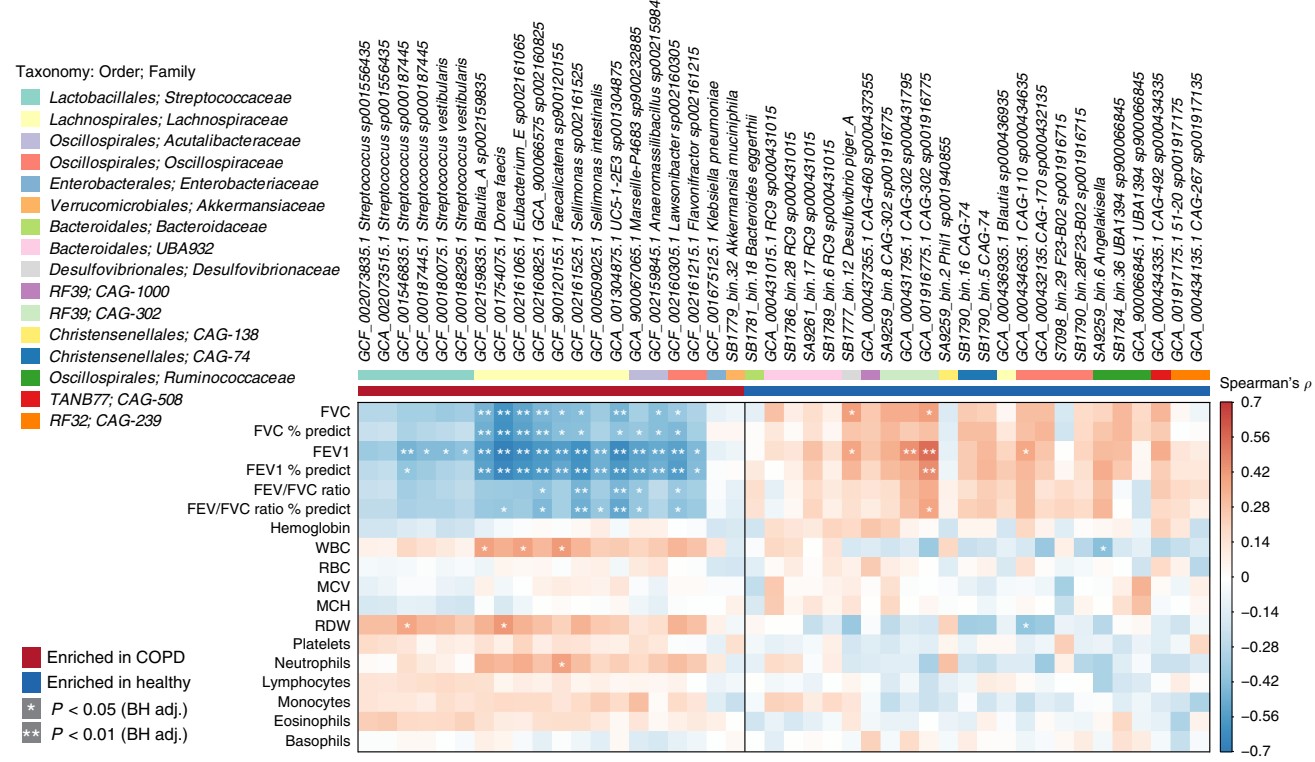

**Fig. 3 Correlation of members of the faecal microbiome with lung function.** Spearman's rho calculated between mapping-based read counts per genome and phenotypic scores. Genomes included are those from Supplementary Data 11, with enrichment in either COPD or healthy samples indicated by the coloured bar along the top of the heatmap. White stars within heatmap boxes indicate significant results (*$p < 0.05$; **$p < 0.01$, Student's t test (two-sided), Benjamini–Hochberg adjustment for multiple comparisons. Exact $p$ values are provided in Supplementary Data 34). Genome abundances were centred with log-ratio transformation prior to analysis. FVC forced vital capacity, FEV forced expiratory volume, WBC white blood cell, RBC red blood cell, MCV mean corpuscular volume, MCH mean corpuscular haemoglobin, RDW red cell distribution width. COPD: $n = 28$; healthy: $n = 29$.

multi-omic signature (Fig. 4 and Supplementary Fig. 3b). Analysis of species confirmed enrichment of *S. parasanguinis_B* and *S. salivarius* in association with COPD (Fig. 4a, b). Within the metabolome, COPD samples were largely defined by depletion of metabolites, with 76% of the identified signature being metabolites present at higher abundance in healthy samples (Fig. 4c, d and Supplementary Data 25). Of the top 50 indicator metabolites separating COPD from healthy samples, 46% were from the lipid ($n = 23$), 20% amino acid ($n = 10$) and 20% xenobiotic ($n = 10$) classes (Supplementary Data 25), indicating that lipid metabolism may be altered in COPD. Sixteen of these compounds, all from the lipid, amino acid or xenobiotic classes, were identified as significantly differential between COPD and healthy samples following adjustment for covariates (age, sex and BMI) using a linear model (Supplementary Data 25). Correlation analysis between the 44 bacterial genomes identified above (Supplementary Data 11) and these 16 metabolites revealed 253 significant associations, many of which involved species enriched in COPD (Fig. 5).

**Lipid involvement in the COPD faecal metabolome.** Within the lipid class, all six metabolites identified as significant in the linear model were enriched in healthy samples (Supplementary Data 25). Four of these were the dicarboxylic acids suberate (C8), sebacate (C10), undecanedioate (C11) and dodecanedioate (C12) that may originate from the diet or be produced endogenously via the ω-oxidation of fatty acids[41,42]. Each of these four lipid metabolites was negatively associated with the majority of species enriched with COPD, suggesting possible 'guilt-by-association' related to the COPD versus healthy divide (Fig. 5). In contrast,

only a subset of species enriched in healthy samples was positively associated with the four dicarboxylates. Bacterial catabolism of dicarboxylic acids has been described in vitro[43]; therefore, we looked for the described enzymes within the genomes of the enriched and depleted species (Supplementary Data 11). While some species are potentially capable of degrading dicarboxylic acids, the pattern of enzyme presence did not match the observed associations with species abundance, either within the healthy or COPD samples (Fig. 5, Supplementary Data 26), supporting a human-derived component of the phenotype. Since the use of statins can influence the rate of fatty acid oxidation[44], we added statin use to the linear model described above. Dicarboxylic acids were no longer significantly depleted in COPD samples following this adjustment (Supplementary Data 25), indicating that statin medication may be driving this phenotype. Inclusion of additional medication (proton-pump inhibitors, selective serotonin-reuptake inhibitors, beta-blockers, angiotensin-converting enzyme inhibitors and angiotensin II receptor antagonists) in an extended linear model reduced the number of significant metabolites to five: amino acid metabolites N-acetylglutamate and N-acetylproline and the xenobiotic metabolites cotinine, asmol and N-carbamoylglutamate, again implicating medication use as impacting the levels of other metabolites.

**Amino acid involvement in the COPD faecal metabolome.** Without adjustment for medication, two amino acid metabolites were enriched and three were depleted in COPD. The first enriched metabolite, N-acetylcadaverine, has previously been associated with Crohn's disease[45]. The precursor of N-acetylcadaverine, cadaverine, is formed during lysine degradation; however, cadaverine levels

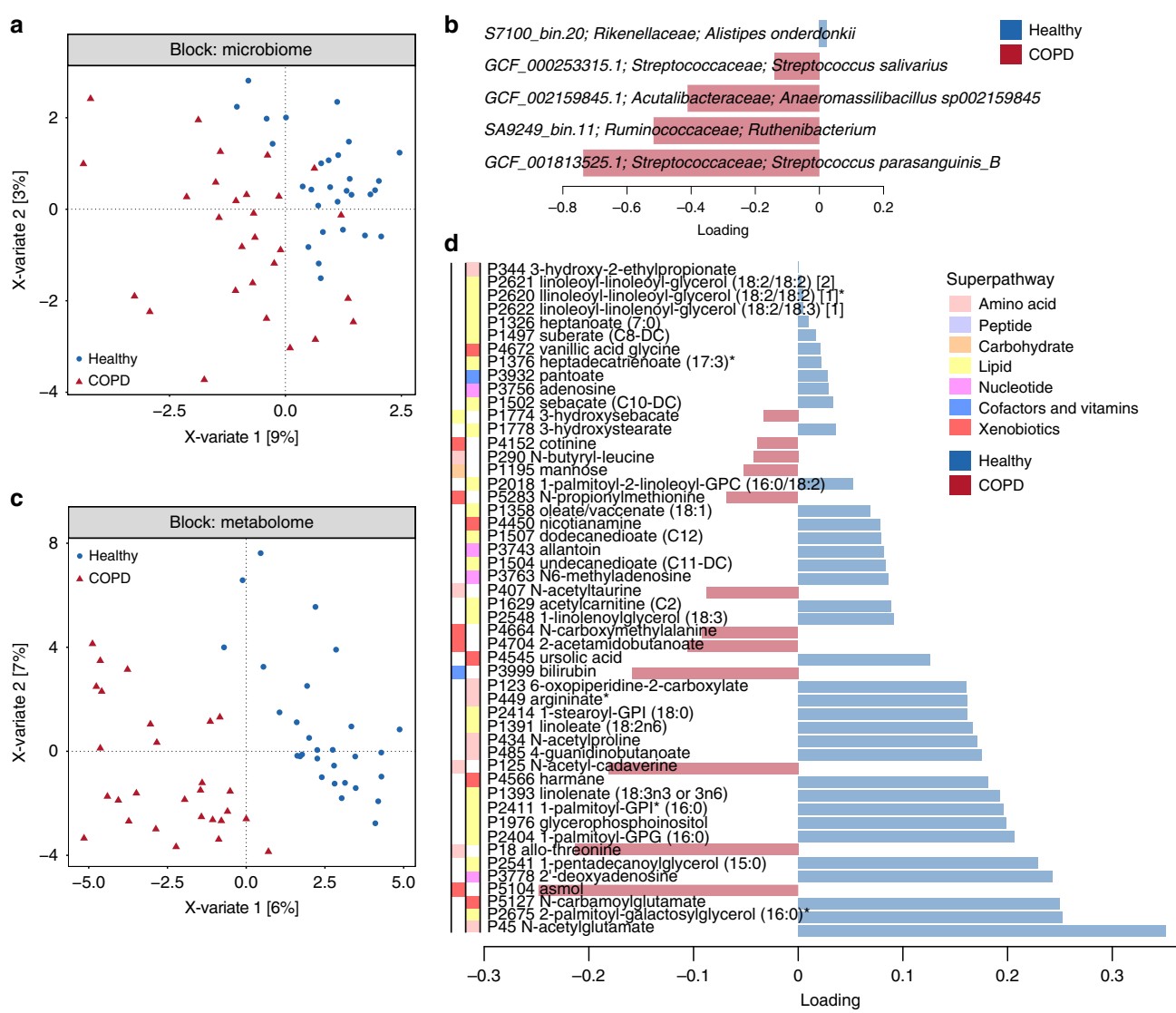

**Fig. 4 Faecal metabolome of COPD patients ($n = 28$) is distinguished from that of healthy individuals ($n = 29$) using a multi-omic analysis. a** DIABLO sample plot demonstrating discrimination between COPD and healthy samples based on microbiome data. **b** Genomes contributing to separation along with component 1 of (**a**). Bar length indicates loading coefficient weight of selected genomes, ranked by importance, bottom to top; bar colour indicates the group in which the sequence variant has the highest median abundance, red = COPD, blue = healthy. Microbiome data are centred log-ratio-transformed relative abundance, filtered for genomes with minimum 0.05% relative abundance in at least ten samples. **c** DIABLO sample plot demonstrating discrimination between groups based on metabolomics data. **d** Metabolites contributing to separation along with component 1 of (**c**). Metabolome data are log-transformed median-scaled values with missing values imputed using the minimum value for each compound, filtered for metabolites returning measurements in at least ten samples.

were not significantly different between COPD and healthy samples (Supplementary Data 25). Microbial production of N-acetylcadaverine has been reported in the soil bacterium *Corynebacterium glutamicum*[46]; however, we did not observe any positive associations between the metabolite and COPD-associated species (Fig. 5), and only one species, *Rothia mucilaginosa_A*, is predicted to carry the N-acetyltransferase required for its production (Supplementary Data 26). The second enriched amino acid metabolite, N-acetyltaurine, can be produced endogenously from taurine; however, there was no significant difference in taurine levels between COPD and healthy samples (Supplementary Data 25). In urine, elevated levels of N-acetyltaurine are used as a marker of ethanol metabolism[47]; however, it is unclear what the biological significance is in faeces. Alcohol consumption was also significantly lower in COPD patients (Supplementary Data 27). The capacity to use N-acetyltaurine as a carbon source has been

described in several marine bacteria[48] and, while we identified homologues of an N-acetyltaurine ABC transporter in the majority of genomes associated with both COPD and healthy samples, only two, *Anaeromassilibacillus sp002159845* and *Lachnospiraceae GCA-900066575 sp002160825*, encoded homologues of the amidohydrolase required for converting N-acetyltaurine to taurine (Supplementary Data 26). Both amidohydrolase-encoding species positively correlated with the abundance of N-acetyltaurine, although they were not the only species displaying this trend (Fig. 5).

Of the three depleted amino acid metabolites in COPD without adjusting for medication, N-acetylglutamate, N-acetylproline and 6-oxopiperidine-2-carboxylate, the first two were also significantly depleted in the extended linear model (Supplementary Data 25). N-acetylglutamate is both a human and microbial-derived metabolite, and may also be ingested[49]. In humans,

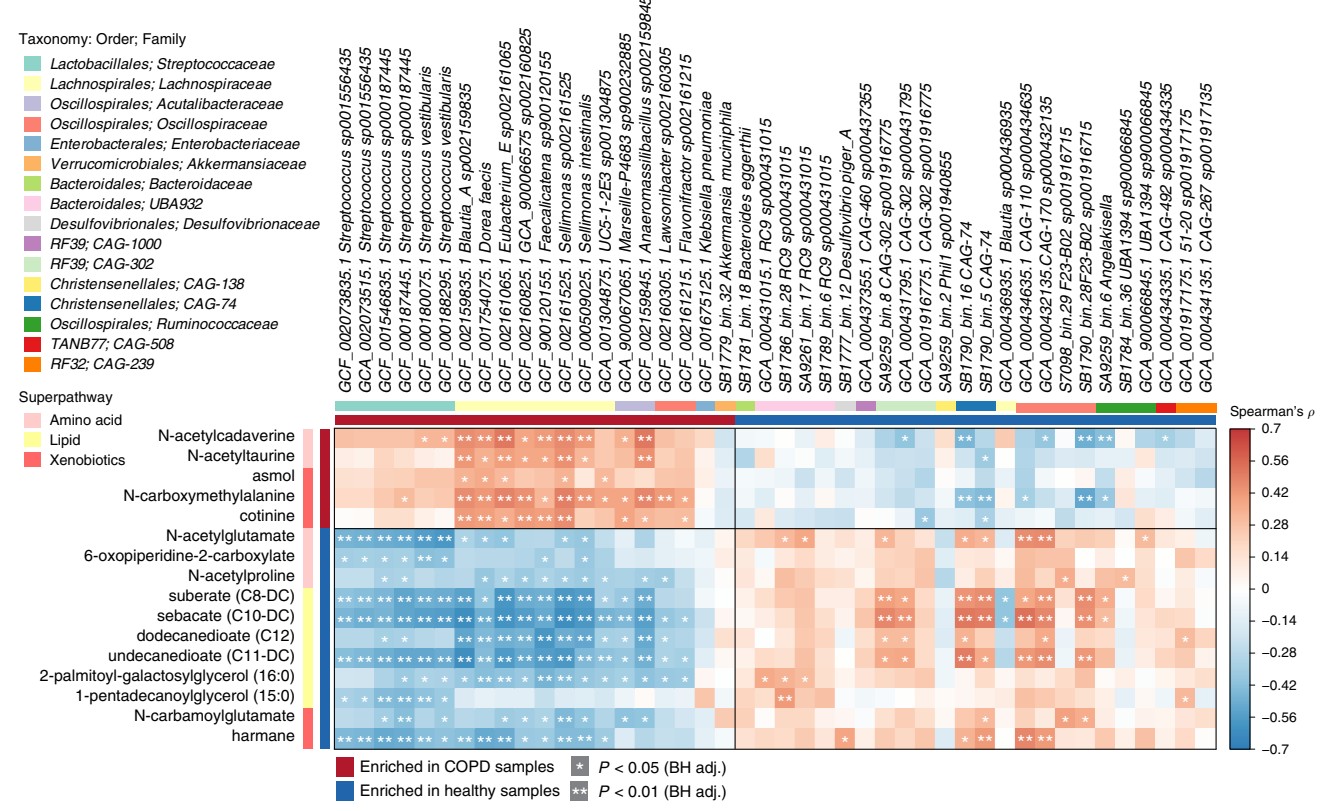

**Fig. 5 COPD-associated species correlate with metabolites differentiating COPD (n = 28) and healthy (n = 29) individuals.** Species and metabolites included are those identified as significantly differential between COPD and healthy samples, including age, sex and BMI within the relevant models (Supplementary Data 7 and 21). Enrichment in either group indicated by coloured bars to the left and top of the plot. Significant correlations denoted by white stars (*p < 0.05; **p < 0.01, Student's t test (two-sided), Benjamini–Hochberg adjustment for multiple comparisons. Exact p values are provided in Supplementary Data 35). Higher taxonomy of species (order, family) and super pathway of metabolites are indicated by coloured bars.

N-acetylglutamate functions as a cofactor for carbamoyl phosphate synthetase I, the first enzyme in the urea cycle, while in bacteria, it is the first intermediate in the arginine biosynthetic pathway[50]. No other elements of the urea cycle were identified as significant (Supplementary Data 25). The majority of genomes enriched in COPD encode N-acetylglutamate synthase, necessary for the generation of N-acetylglutamate from glutamate, versus five of the genomes enriched in healthy samples (Supplementary Data 26). This suggests that the increased abundance of the metabolite in healthy samples may be a product of endogenous metabolism or altered dietary intake. The role of the other two amino acid metabolites enriched in healthy samples is unclear. N-acetylproline has been associated with the consumption of processed protein[51] and may therefore relate to diet. 6-oxopiperidine-2-carboxylate is a by-product of penicillin production by *Penicillium chrysogenum*[52].

**Xenobiotic involvement in the COPD faecal metabolome**. Within the xenobiotic class, metabolites increased in COPD include the tobacco metabolite cotinine and the respiratory drug salbutamol (asmol), the usage of which was reported by 70% (n = 20) of patients (Supplementary Data 2). Both cotinine and salbutamol remained significant in the extended linear model (Supplementary Data 25). Depleted xenobiotic metabolites, N-carbamoylglutamate and harmane, both have potential beneficial effects in the gut. N-carbamoylglutamate is an analogue of N-acetylglutamate and has beneficial roles in the animal gut following supplementation, including stimulating arginine synthesis[53], protection against oxidative stress[54] and epithelial cell

proliferation[55]. However, its source within the human gut is unknown. The β-carboline alkaloid harmane is found in plants and is also a bacterial metabolite[56] and may therefore have multiple origins in the gut. Harmane has antimicrobial properties[57] and may modulate the innate immune system[58].

**A disease-associated network in COPD**. We also undertook network analysis based on the integration of metabolomics and metagenomic datasets using species and metabolites identified in >10 samples, as described above (Fig. 4), to look for associations between the broader microbiome and COPD-linked metabolites. Three distinct microbiome/metabolite clusters were defined (Fig. 6). The first indicated associations between *S. parasanguinis_B*, *Ruthenibacterium* sp. and *Anaeromassilibacillus sp002159845* and a group of 13 metabolites (Fig. 6a), each identified as different between COPD and healthy samples in our multivariate analysis (Fig. 4d). Eight of these were also discriminatory following adjustment for age, sex and BMI; one enriched and seven depleted in COPD (Supplementary Data 25). The second and third networks do not contain any nodes enriched in COPD or healthy samples and therefore likely represent interactions additional to a disease state (Fig. 6b, c). The first cluster, therefore, represents a shortlist of disease-associated species and metabolites for future testing in clinical models.

**Enrichment of *Streptococcaceae* family members in the COPD-associated gut microbiome is replicated in an independent validation cohort**. To validate our microbiome findings, we undertook metagenomic sequencing of a validation cohort

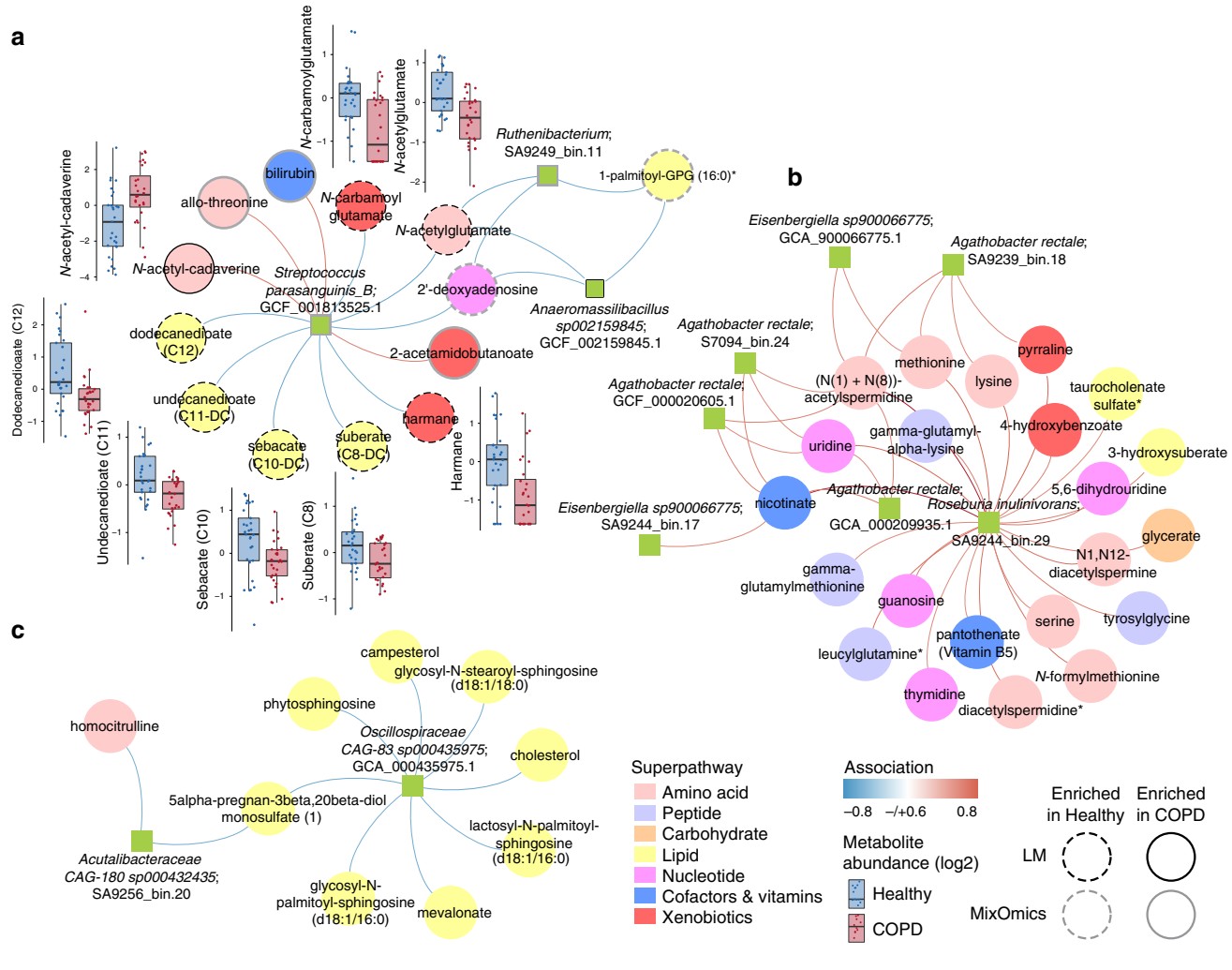

**Fig. 6 Integration of faecal microbiomes and metabolomes identifies a COPD-associated network. a–c** Integration of microbiome and metabolome datasets using the software DIABLO produced association networks showing correlations between bacterial species and metabolites. A positive correlation between nodes is indicated by red connecting lines, negative correlation by blue. Species and metabolites enriched in COPD or healthy samples are denoted by solid or dashed borders, respectively. Black borders indicate significance in the linear model adjusted for age, sex and BMI ($p < 0.05$, Wald test (two-sided) with Benjamini–Hochberg adjustment for multiple comparisons, Supplementary Data 7 and 21) and grey borders indicate selection by MixOmics as discriminatory along with component 1 of Fig. 4a, b. All metabolites significant within the linear model were also selected by MixOmics. The abundance of metabolites (log-transformed median scaled) significant in the linear model provided as boxplots adjacent to the relevant nodes. Each box centres on the median, with lower and upper bounds representing the first and third quartile (25th and 75th percentile), respectively. Whiskers extend 1.5 times the interquartile range from the outer bounds. Microbiome data filtered for genomes with minimum of 0.05% relative abundance in ≥10 samples. Metabolome data filtered for metabolites returning measurements in ≥10 samples. Microbiome data are centred log-ratio- transformed relative abundance. Metabolomics data are log-transformed median-scaled values with missing values imputed using a minimum value for each compound. COPD: $n = 28$; healthy: $n = 29$.

comprising 38 samples, 16 COPD patients and 22 healthy individuals (Supplementary Data 31). As with the study cohort, COPD and healthy stool samples could be distinguished based on bacterial community profiles ($p = 0.037$, PERMANOVA of Bray–Curtis distances), with COPD status explaining ~4% of between-sample variability (compared to 6% in the study cohort). Of the 210 genomes identified as enriched in either COPD or healthy samples in the study cohort (Supplementary Data 10), 59 (28%) displayed a similar enrichment trend in the validation cohort of which 33 (16%) reached significance including six of the *Streptococcus* spp. enriched in COPD samples, and *RC9* spp., *CAG-302* spp. and *UBA11524 sp000437595* enriched in healthy samples (Supplementary Data 32). Using a multivariate approach,

11 (37%) of the 30 genomes identified as key differentiators of COPD and healthy samples in the study cohort were in the top 30 separating the groups in the validation cohort (Fig. 7a, b). Along with *Streptococcus parasanguinis_B*, highlighted in the disease-associated network (Fig. 6), these species included *Eubacterium_E sp002161065*, *Sellimonas* spp., *Anaeromassilibacillus sp002159845* and *Lawsonibacter sp002160305* that correlated with lung function in the study cohort (Fig. 3). At the functional level, six of the eight domains significantly enriched in COPD samples (Supplementary Data 8–11) followed a similar trend in the validation cohort, although none significantly so (Supplementary Data 33). This indicates that larger cohorts may be required to clearly differentiate COPD samples based on gut

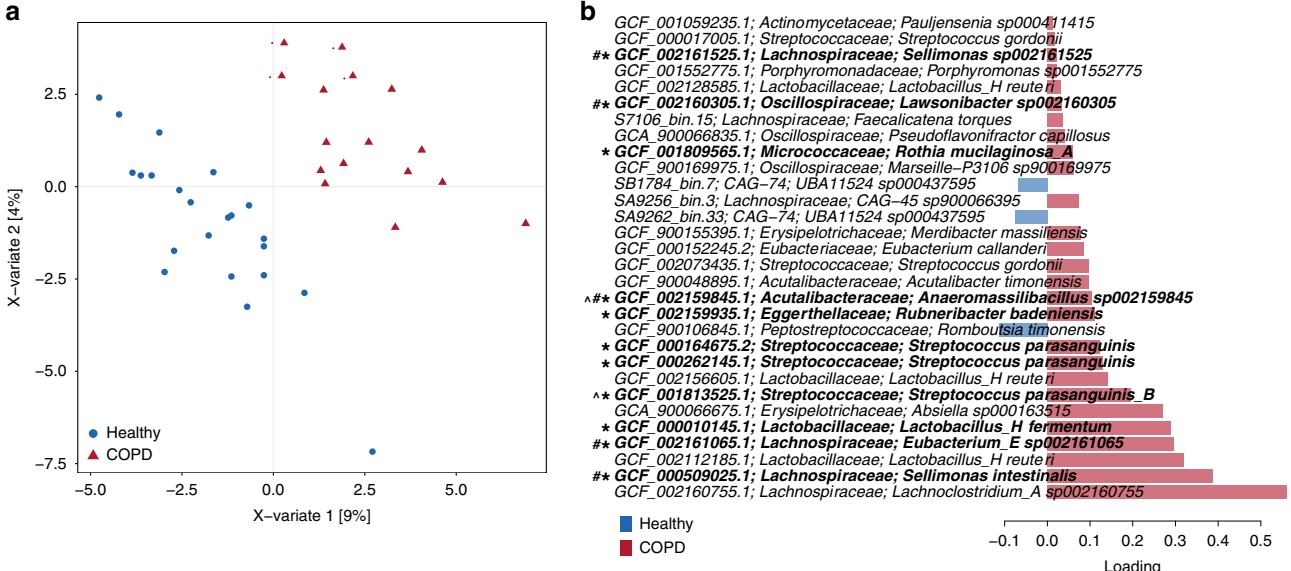

**Fig. 7 Association of gut microbiome members with COPD replicate in an independent cohort. a** Multivariate sparse partial least-squares discriminant analysis (sPLS-DA) of read-mapping-based relative abundance at the genome level of the faecal microbiome, filtered for genomes with minimum 0.05% relative abundance in at least one sample. **b** Genomes contributing to separation along with component 1 of sPLS-DA from (**a**). Bar length indicates loading coefficient weight of selected genomes, ranked by importance, bottom to top; bar colour indicates the group in which the genome has the highest median abundance, red = COPD, blue = healthy. Genomes marked with * are those within the discriminatory signature defined for the study cohort (Fig. 2), # indicates genomes associated with clinical phenotypes (Fig. 3) and ^ indicates genomes within the disease-associated network (Fig. 6). COPD: n = 16; healthy: n = 22.

metagenome functional capacity. These data do, however, validate the association of specific members of the gut microbiome with COPD, providing further impetus for their testing in disease models.

## Discussion

We present the first analysis of the human gut microbiome and metabolome in COPD to complement previous work focused on the lung. We reveal that both the faecal microbiome and metabolome of stable COPD patients are significantly different from that of healthy controls. There was no difference in microbiome composition between current smokers compared to non-smokers with COPD, supporting this as a disease-associated phenotype rather than one driven by the influence of cigarette smoke on the gut microbiome[59]. Several elements of the newly described COPD gut metabolome suggest altered systemic metabolism associated with the disease, the outcomes of which are detectable in faecal samples promoting faecal sampling as a means of monitoring disease. Since changes in metagenomes correlated with disease features, the processes involved may have the potential to be therapeutic targets or the outputs used as faecal biomarkers, although this would need clinical and experimental validation.

We found increased abundance of several *Streptococcus* species, including *S. parasanguinis_B* and *S. salivarius* in COPD, which was partially replicated in an independent validation cohort. *Streptococcus* enrichment was associated with increased abundance of glucosyltransferase and LPXTG-anchored adhesion domains, suggesting that adhesive capacity was key to increased abundance. Streptococci are pioneer colonisers and some of the first species detected in the oral cavity and gut of infants[28]. Increased abundance of *Streptococcus* in the gut has been observed in association with smoking[60], and several studies of the lung microbiome of COPD patients have also noted an increased abundance of the genus[13,14]. *S. parasanguinis_B* was also isolated from the sputum of a COPD patient experiencing an acute exacerbation (GCF_000963275.1)[61]. One possible explanation for

the presence of these organisms in both the lung and gut is a transfer from the oral microbiome. *Streptococcus* strains exhibit frequent oral–faecal transmission in healthy adults[62], and transmission rates may increase in COPD where microaspiration of the airways with pharyngeal secretions is exaggerated[63]. Increased *Streptococcus* across distinct mucosal niches in addition to the non-uniform progression to COPD amongst smokers[6], also supports a potential genetic predisposition associated with this phenomenon, such as altered mucosal immunity[64] or antibody secretion[65], although twin-based analysis suggests environment rather than genotype as the primary explanatory variable in oral streptococci abundance[66].

While streptococci were associated with COPD status, we found a limited correlation between *Streptococcus* species abundance and lung function and no correlation with other disease metrics. Multiple members of the family *Lachnospiraceae* were correlated with reduced lung function. *Lachnospiraceae* members have been associated with both healthy[67,68] and disease-associated[69,70] gut microbiomes, and a subset of *Dorea* species has also been associated with the release of inflammatory cytokines[71]. Contrasting phenotypic effects within genera highlight the interspecies variability that complicates microbiome data interpretation and prevents extrapolation to uncharacterised species such as those described here. Further work is required to determine whether the identified species are actively contributing to the established relationship between airway neutrophilia and lung function decline in COPD[72], or whether they are responding to altered conditions independently associated with the disease.

Two metabolites reduced in COPD patients are cofactors of carbamoyl phosphate synthetase I, the first enzyme in the urea cycle, the native cofactor, N-acetylglutamate and its structural analogue, N-carbamoylglutamate. N-carbamoylglutamate has been characterised in the livestock industry due to its capacity to stimulate arginine synthesis[53]. Arginine is an important mediator of gut health[73] and also contributes to airway function[74]. We found no difference in the concentration of arginine (or other

urea cycle intermediates) between COPD and healthy individuals. However, analysis of BALF from patients identified a negative association between several amino acids, including arginine and lung function[75], suggesting that there may be a systemic effect of reduced cofactor levels that does not appear in the faeces. N-carbamoylglutamate has also been associated with omega-3 fatty acid intake in humans, and a possible link between bacterial production of N-carbamoylglutamate and fatty acids has been suggested[76]; however, it is currently unknown which bacteria may be producing the compound.

We also observed reduced levels of dicarboxylic acids in COPD patients, potentially driven by increased statin use within the cohort. These metabolites are generated endogenously via the omega-oxidation of fatty acids and are excreted in the urine, with increased levels associated with a number of diseases[77]. Two of the dicarboxylic acids identified as depleted, suberate (C8) and sebacate (C10), along with azelate (C9), were identified as positively associated with FEV based on serum analysis, however, were not significantly associated with a diagnosis of COPD[78]. Statin use was not reported in that study. Impaired fatty acid metabolism has been indicated in COPD based on reduced fatty acid oxidation by isolated peripheral blood mononuclear cells from patients compared to those from healthy smokers[79]. Reduced levels of β-oxidation in female, but not male, COPD patients are also suggested based on serum analysis[80]. A shift in lipid metabolism may therefore still be associated with COPD; however, it may require a larger cohort to tease apart from the influence of medication. A decrease in dicarboxylic acids has also been observed in association with inflammatory bowel disease[81,82]; however, it is possible that medication profiles also affect these outcomes.

Interestingly, we observed a lower dietary fibre intake in participants with COPD compared to controls based on dietary surveys, which may contribute to both differences in gut microbiome profile and COPD pathology. Dietary fibre resists digestion in the small intestine and upon reaching the colon, soluble forms are partially fermented by commensal bacteria. Some soluble fibres act as prebiotics, providing a selective growth substrate, leading to changes in bacterial number and diversity and increased production of immunosuppressive by-products[21], which have been shown to reduce airway inflammation in both animal[83] and humans[84,85] models of asthma. Hence, increasing fibre intake in COPD may be a relevant therapeutic strategy, as previously suggested[26].

Analysis and integration of omic datasets are challenging due to the many variables that can influence associations, resulting in a suboptimal rate of validation in the laboratory[82]. Here we attempt to confirm observed microbe–metabolite associations using the encoded genetic potential of the species in question, focusing on species and compounds identified as distinct between COPD and healthy samples. Although we observed overlap in genetic potential, we did not find a clear connection between the datasets. While this may be due to the action of external factors, notably medication, it is also possible that the species responsible for the metabolite signature are not differentially abundant between groups. Rather, differential activity levels, triggered by disease-specific environmental variables and uncaptured by inferred metabolic potential, may induce the signature. To assess this, complementary *meta*-transcriptomic or proteomic analyses of the microbiome are needed and may yield improved integration of microbial and metabolomic datasets.

Recognised variation in gut microbiome profiles between individuals and confounders such as medication status likely limited our ability to detect additional significant taxonomic and functional biomarkers for COPD. However, encouragingly there was a significant overlap between our relatively small study and

validation cohorts. Analysis of larger COPD cohorts will likely identify additional significant correlated biomarkers. Our study was also limited to steady-state disease and therefore did not capture the gut environment during disease exacerbation. Longitudinal analysis during exacerbation and recovery would be particularly interesting if paired with a similar sampling of the lung environment to evaluate potential seeding from the gut. A design incorporating such repeated sampling of the same individual would also help overcome the problem of inter-individual variation.

Despite these limitations, a discriminatory signal is present in both the metagenomic and metabolomic datasets supporting the gut as a potential source of disease biomarkers in COPD. These candidates should be further evaluated for their mechanistic and causal involvement in COPD using established animal models[7,86,87].

## Methods

**Patient characteristics.** Twenty-eight COPD patients and 29 healthy controls were recruited from John Hunter Hospital, Belmont District Hospital, Newcastle Community Health Centre, Westlakes Community Health centre and Hunter Medical Research Institute (Newcastle, Australia). All participants provided written informed consent, and ethics approval was obtained from the Human Ethics Research Committees of the Hunter New England Local Health District (14/08/20/3.02) and the University of Newcastle (H-2015-0006). COPD was defined by the GOLD standard of post-bronchodilator $FEV_1 < 80\%$ predicted and $FEV_1/FVC < 0.7$, and by physician diagnosis; all were >40 years old and had a previous history of smoking. Healthy controls were adults >40 years old with no history of cardiac or respiratory disease, and with normal lung function measured by spirometry ($FEV_1/FVC$ ratio >0.7 and FEV1 > 80% predicted). Participants were excluded if they had received treatment with an antibiotic or oral prednisone, experienced significant abdominal pain, bloating, diarrhoea or respiratory tract infection in the previous 4 weeks, or had a previous history of gastrointestinal disease. Current and ex-smokers were not excluded.

For the validation cohort, 16 COPD patients and 22 healthy participants were recruited through the thoracic outpatient clinic at The Prince Charles Hospital and the general population, respectively. All participants provided written informed consent, and ethics approval was obtained from The Prince Charles Hospital Human Research Ethics Committee (HREC/18/QPCH/234) and the University of Queensland (2108001673/HREC/18/QPCH/234). Patients were included in the study if they had COPD as defined by the GOLD guidelines (chronic airflow limitation that is not fully reversible, with post-bronchodilator FEV1/FVC < 70% and FEV1 < 80% predicted). COPD patients were former smokers of ≥10 years, who are recruited during stability (>4 weeks since an exacerbation). Healthy controls were adults >40 years old with no history of cardiac or respiratory disease. Participants were excluded from the study due to any antibiotic or oral corticosteroid use in the past 4 weeks, a current smoker, had comorbid lung disease (e.g. asthma, lung cancer, interstitial lung disease and bronchiectasis) that interferes with the study outcomes, had other co-morbidities with established altered microbiome (including IBD, irritable bowel syndrome), or extreme dietary habits that may significantly impact gut microbiome composition.

Statistical comparison of metadata characteristics between COPD and healthy groups (Supplementary Data 1 and 27) was undertaken in R using either Student's *t* test (two-sided) or Wilcoxon rank-sum test dependent on normality estimation using Shapiro–Wilk test. Pearson's chi-squared test was used for categorical variables. Comparison of dietary questionnaire responses was undertaken using a Wilcoxon rank-sum test with Benjamini–Hochberg adjustment for multiple comparisons.

**Specimen collection.** Individuals who consented to participate were first screened via phone interview, and suitable candidates attended the Hunter Medical Research Institute for a formal assessment. Individual history was recorded, including symptoms, medical and medication history, smoking history and completion of a Dietary Questionnaire for Epidemiological Studies (Version 2, Cancer Council Victoria, Australia). For COPD patients, a history of exacerbations in the last 12 months was also recorded and health status measured using the COPD assessment tool. Spirometry (Easyone) was performed post bronchodilator to assess airway obstruction and a plasma sample collected and stored at −80 °C. Participants were supplied with a faecal collection kit and instructed to collect faeces within 48 h of their visit. Faecal samples were stored in the participants' freezer until returned frozen for analysis. Samples were stored at −80 °C until processed.

**DNA extraction and sequencing.** DNA was extracted from ~100 mg of faecal material using an initial bead-beating step followed by extraction using a Maxwell 16 Research Instrument (Promega, USA) according to the manufacturer's protocol

with the Maxwell 16 Tissue DNA Kit (Promega, USA). DNA concentration was measured using a Qubit assay (Life Technologies, USA) and was adjusted to a concentration of 5 ng/μl. The 16S rRNA gene encompassing the V6–V8 regions were targeted using the 803 F (5′-TTAGAKACCCBNGTAGTC-3′) and 1392 R (5′-ACGGGCGGTGWGTRC-3′) primers modified to contain Illumina specific adaptor sequences (803F:5′TCGTCGGCAGCGTCAGATGTGTATAAGAGACAG TTAGAKACCCBNGTAGTC3′ and 1392wR:5′GTCTCGTGGGCTCGGGTC TCGTGGGCTCGGAGATGTGTATAAGAGACAGACGGGCGGTGWGTRC3′). Library preparation was performed as described, using the workflow outlined by Illumina (#15044223 Rev.B). In the first stage, PCR products of ~590 bp were amplified according to the specified workflow with an alteration in polymerase used to substitute Q5 Hot Start High-Fidelity 2X Master Mix (New England Biolabs, USA) in standard PCR conditions. The resulting PCR amplicons were purified using Agencourt AMPure XP beads (Beckman Coulter, USA). Purified DNA was indexed with unique 8-bp barcodes using Illumina Nextera XT 384 sample Index Kits A–D (#FC-131-1002, Illumina, USA). Indexed amplicons were pooled in equimolar concentrations and sequenced on the MiSeq Sequencing System (Illumina, USA) using paired-end sequencing with V3 300 bp according to the manufacturer's protocol. Metagenomic sequencing was performed using the same DNA extractions. Library preparation was performed using the Nextera DNA Library Preparation Kit (Illumina, USA). Libraries were sequenced using the Illumina NextSeq500 platform generating approximately 2 Gbp of 150-bp paired-end reads per sample. Metagenomic sequencing of the validation cohort was undertaken by Microba (Brisbane, Australia) generating approximately 6 Gbp of 150-bp paired-end reads per sample.

**16S rRNA gene sequencing analysis**. Reads were cleaned of adaptor sequences using Cutadapt v1.1[88] and trimmed using Trimmomatic v0.36[89] employing a sliding window of 4 bases with an average base quality above 15, followed by hard trimming to 250 bases with the exclusion of reads less than this length. Read statistics are provided in Supplementary Data 28. The remaining forward reads were processed following the QIIME2 workflow[90] using DADA2 v1.12[91] to denoise sequences. Taxonomy assignment was performed on amplicon sequence variants using BLAST v2.8.1[92] against the SILVA[93] reference database version 132. Read counts were normalised prior to PCA and heatmap visualisation using log-transformed cumulative-sum scaling implemented within metagenomeSeq v1.24.1[94]. PCA was performed using the rda function and PERMANOVA using the adonis function within the vegan v2.5-5R package[95]. Heatmaps were generated using the heatmap v1.0.12R package[96]. Alpha-diversity was calculated using QIIME v1.8.0[90] with raw, unfiltered counts. sPLS-DA analysis was conducted using the R package mixOmics v6.6.2[31] using log-transformed cumulative-sum-scaled values with $10 \times 10$-fold cross-validation, including sequence variants present at ≥0.05% relative abundance in ≥3 samples.

**Metagenomic sequence processing and recovery of MAGs**. Contaminating human reads were identified by mapping against the human genome (Homo_-sapiens.GRCh38, https://www.ncbi.nlm.nih.gov/assembly/2334371) using BWA v0.7.12[97] requiring a minimum alignment length of 30 bases and maximum of 15 clipped bases for reads to be considered of human origin. Adaptor removal and read trimming were performed using Trimmomatic v0.36[89] with the following settings: LEADING:3 TRAILING:3 SLIDINGWINDOW:4:15 MINLEN:50. Read statistics are provided in Supplementary Data 29. Each sample was assembled independently using Spades v3.12.0[98] with the –meta flag. Reads were mapped to each resulting assembly using BamM v1.7.3 (https://github.com/ecogenomics/BamM) and bins produced using Metabat v2.12.1[99] with a minimum contig length of 1500 bases. Contamination and completeness of bins from all samples were assessed using CheckM v1.0.11[100]. Bins with completeness >80% and contamination <7% were retained and de-replicated using dRep v2.05[101] with default settings (99% identity), skipping quality filtering. The taxonomic affiliation of recovered MAGs was determined using the Genome Taxonomy Database (GTDB) Releases 03-RS86 and 04-R89[102] using GTDB-Tk v0.3.0[103] (Supplementary Data 30).

**Metagenomic community profiling**. Reads for each sample were mapped to a de-replicated set of 23,936 genomes from NCBI (GTDB Release 03-RS86)[102] using BamM with minimum seed length of 25. Genomes with >1× coverage or >1% of the genome, as determined using Mosdepth v0.2.3[104], were retained ($n = 1229$, Source Data) and combined with study MAGs for assessment of community composition. dRep[101] was used to identify overlap (99% identity) between study MAGs and NCBI genomes, where overlap occurred, MAG was retained. Read counts for the final genome set were determined for each sample via mapping using BamM with minimum seed length of 25 bases and subsequent filtering for minimum mapping percentage identity of 95%. Per-genome read counts were scaled to account for genome size whilst maintaining the raw unmapped read percentage for each sample as a reflection of unrepresented diversity. Relative abundance was calculated using scaled read counts as a fraction of total non-host reads per sample. Alpha-diversity was calculated using QIIME v1.8.0[90] with counts normalised using the size-factor method implemented within the R package DESeq2 v1.22.2[30].

PCA was conducted using the R package vegan v2.5-1[95] on data normalised using log-cumulative-sum scaling (log-CSS) implemented within metagenomeSeq

v1.22.0[94]. Differential abundance of bacterial taxa between groups was assessed using the Wald test within DESeq2 v1.20.0[30] based on read counts scaled to account for genome size with the Benjamini–Hochberg adjustment for multiple comparisons. The genome-level analysis was conducted using genomes present with at least 0.05% relative abundance in one sample. sPLS-DA analysis was conducted using the R package mixOmics v6.6.2[31] using centred log-ratio-transformed relative abundance with $50 \times 15$-fold cross-validation. Correlation analysis between metagenomic and phenotypic data was undertaken using genomes identified as significantly different between COPD and healthy samples following removal of patient confounders (Supplementary Data 11). Spearman's rho was calculated using 'corr.test' function within R package psych v1.8.12[105] based on centred log-ratio-transformed genome relative abundance. A correlation matrix was produced using 'corrplot' function with R package corrplot v0.84[106].

**Metagenomic functional profiling**. For read-based analysis, protein fragments in raw reads were predicted using Prodigal v2.6.3[107] and subsequently alignment with HMMER v3.1b2[108] to the hidden Markov model databases dbCAN CAZy v6[109], Pfam r31[110] and TIGRFAM v15[111] with a maximum e-value cut-off of 1e−10. KEGG orthology was determined via BLAST v2.8.1[92] alignment to UniProt Uni-Ref100 database downloaded on July 2017[112] with maximum e value of 1e−10 and subsequent extraction of associated KO terms. Counts per sample were used to compare group functional profiles with DESeq2 v1.20.0[30] following removal of domains with total read counts ≤10% of the average read count across all domains. Genome-level analysis of KEGG orthology terms and module completeness was undertaken using EnrichM v0.5.0 (https://github.com/geronimp/enrichM) with maximum e value of 1e−10 and Fisher's exact test with Benjamini–Hochberg adjustment used to assess significance. Comparison of module completeness was undertaken in R using the Wilcoxon rank-sum test with Benjamini–Hochberg adjustment. The presence of genes of interest (i.e. related to an enriched meta-bolite) in enriched genomes was determined using BLAST with minimum e value 1e−10, identity 30% and alignment length 70%. Protein sequences used as queries are included in Supplementary Data 26.

**Metabolite extraction, profiling and analysis**. Metabolites were profiled in faecal samples by Metabolon Inc. (Durham, NC, USA). All samples were maintained at −80 °C until processed as previously described[113]. Global metabolic profiles were determined using the Metabolon HD4 platform. Samples were prepared using the automated MicroLab STAR® system (Hamilton Company, USA), with several recovery standards added prior to extraction and processing for quality control. To recover chemically diverse metabolites and precipitate protein and dissociate small molecules bound to protein in the precipitated matrix, samples were extracted with methanol with vigorous shaking for 2 min (Glen Mills GenoGrinder 2000, USA) followed by centrifugation. The extract was divided into five different fractions for further analysis. The organic solvent was removed by placing briefly on a Tur-boVap® Concentration Evaporator (Zymark). Samples were stored overnight under nitrogen.

The process of ultra performance liquid chromatography (UPLC)/mass spectrometry (MS)/MS was performed with a Waters ACQUITY (UPLC), Thermo Scientific Q-Exactive high-resolution mass spectrometer interfaced with a heated electrospray ionisation (HESI-II) source and Orbitrap mass analyser operated at 35,000 mass resolution. Sample extracts were processed dry and reconstituted to consist of a series of standards at fixed concentrations to have injection and chromatography consistency before detailed analysis with four methods. For more hydrophilic compounds, optimised reverse-phase UPLC–MS/MS with acidic conditions and positive ion-mode electrospray ionisation was used. Here, a C18 column (Waters UPLC BEH C18-2.1 × 100 mm, 1.7 μm), consisting of perfluoropentanoic acid (0.05%) and formic acid (0.1%) was used to gradient-elute the extract using water and methanol. For hydrophobic compounds, extracts were gradient-eluted with the same C18 column using methanol, acetonitrile, water, perfluoropentanoic acid (0.05%) and formic acid (0.01%). Higher organic content was maintained during processing. Basic negative-ion conditions using a separate C18 column were used to elute the basic extract with methanol and water, ammonium bicarbonate (6.5 mM, pH 8). Negative-ion-mode electrospray ionisation conditions with hydrophilic interaction chromatography were used with a Waters UPLC BEH Amide 2.1 × 150-mm, 1.7-μm column. Here, extracts were gradient-eluted with water and acetonitrile with ammonium formate (10 mM, pH 10.8). The mass spectrometry analysis alternated between MS and data-dependent MS$^n$ scans, with scan range covering from (70 to 1000 m/z) achieved with the dynamic elusion method[114].

Metabolon's hardware and software systems were based on LAN backbone; database servers operating on Oracle 10.2.0.1 Enterprise Edition, are utilised to extract, peak-identify and quality-check and process the raw data files. Compound identification is achieved by comparison with library entries of purified standards (or recurrent unknown entities), which consist of retention time/index, the mass-to-charge ratio (m/z) and chromatographic data, including MS/MS spectral data information. Biochemical identification follows the retention time/index window of the proposed identification mass match to the library (±10 ppm) and MS/MS forward and reverse scores. Quality check and curation procedures are followed to ensure that library matches for each compound from each sample are correct.

Peaks are quantified using area-under-the-curve detector ion counts and corrected across multiple runs by adjusting the median value of each compound to 1.

Following median scaling, then imputation of missing values, if any, with the minimum observed value for each compound, the data were transformed to the natural log for statistical analysis. Linear regression of metabolite data was performed using lm package in R implemented within NormalizeMets[115] v0.25[115] incorporating sample group, age, BMI and sex and non-COPD medications within the model matrix as indicated.

**Metabolomic and metagenomic data integration**. Correlation analysis between metagenomic and metabolomic data was undertaken using genomes and metabolites identified as significantly different between COPD and healthy samples incorporating adjustment for age, sex and BMI (Supplementary Data 7 and 21). Spearman's rho was calculated using 'corr.test' function within R package psych v1.8.12[105] based on centred log-ratio- transformed genome relative abundance and log-transformed raw metabolite values. The pseudo count used for each dataset was one order of magnitude below the lowest non-zero value. The correlation matrix was produced using 'corrplot' function with R package corrplot v0.84[106].

DIABLO from the R package mixOmics v6.6.2[31] was used to generate integrated metagenomic and metabolomic signature. The analysis was performed using centred log-ratio-transformed taxa relative abundance (with a pseudo count of 1e−08, one order of magnitude below the lowest non-zero value) and log-transformed median-scaled metabolite data. Taxa were filtered for those present at a minimum of 0.05% in at least ten samples (genome level) and metabolites for those detected in at least 10 samples. The block link within the design matrix was set at 0.1. The optimum number of components and variables included within the final model was determined using the 'tune.block.splsda' function with 50 × 10-fold cross-validation.

**Reporting summary**. Further information on research design is available in the Nature Research Reporting Summary linked to this article.

## Data availability

The 16S rRNA amplicon and metagenomic sequencing data have been deposited to the NCBI Sequence Read Archive under accession PRJNA562766. Recovered MAGs have been deposited to the NCBI DDBJ/ENA/GenBank database under accessions WGSA00000000–WHIU00000000. Prokka annotated MAG sequences in GenBank format are available at https://github.com/katebowerman/COPD. Sample accessions are provided in Supplementary Data 28–30. Sequence variant read counts from 16S rRNA amplicon sequencing (raw data underlying Fig. 1) and metagenomic genome-based mapping counts (raw data underlying Figs. 2–7) are provided as a Source Data File. The reference human genome used in this study (Homo_sapiens.GRCh38) is available at https://www.ncbi.nlm.nih.gov/assembly/2334371. Reference bacterial genomes are available from https://www.ncbi.nlm.nih.gov/assembly/. Additional databases used in this study are available as follows: SILVA v132, GTDB 03-RS86 and 04-R89, dbCAN v6, Pfam r31, TIGRFAM v15 and UniProt UniRef100. Source data are provided with this paper.

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

## Acknowledgements
The authors thank Professor Graham Giles of the Cancer Epidemiology Centre of The Cancer Council Victoria, for permission to use the Dietary Questionnaire for Epidemiological Studies (Version 2), Melbourne: The Cancer Council Victoria, 1996. We thank Lorissa Hopkins and Jasmine Wark for assistance with recruiting patients, collection of human data and samples. This work was funded by grants from the Rainbow Foundation to P.M.H., the National Health and Medical Research Council (NHMRC, 1059238) of Australia to P.M.H. and P.H. and Prince Charles Hospital Foundation Innovations Grants to I.A.Y. (INN2018-30) and A.V. (INN2019-24). P.M.H. is funded by fellowships from the NHMRC (1079187 and 1175134) and A.V. by a fellowship from the Prince Charles Hospital Foundation (RF2017-05).

## Author contributions
Study design (S.L.G., K.F.B., P.A.W., P.H. and P.M.H.), questionnaire selection (S.L.G., K.F.B. and L.G.W.), data collection (S.F.R. and K.F.B.), sample collection (P.A.W., A.V. and I.A.Y.), sample processing (N.L., S.L.G., K.F.B., S.F.R. and S.D.S.), data processing (K.L.B., S.F.R. and D.L.A.W.), data analysis (K.L.B. and S.F.R.), paper preparation (K.L.B., S.F.R., L.G.W., P.H. and P.M.H.) and paper editing and review (K.L.B., S.F.R., A.V., N.L., K.F.B., R.Y.K., D.L.A.W., S.D.S., L.G.W., I.A.Y., P.A.W., P.H. and P.H.M.).

## Competing interests
P.H. is a co-founder of Microba Life Sciences Limited, and D.L.A.W. is currently an employee of Microba. The remaining authors declare no competing interests.
