## [Peer Review File · Nature Communications]

Reviewers' comments:

Reviewer #1 (Gut immunity, metabolite)(Remarks to the Author):

The authors examined 16S rRNA-based microbiome plus metagenomics and metabolome changes in COPD patients. Both microbiome and metabolomics analyses in COPD are fairly well established in the field. A uniqueness of this study is to identify the increase of N-acetylglutamate in COPD patients. The authors emphasize that *Streptococcus parasanguinis*, N-acetylglutamate and its analogue N-carbamoylglutamate could be used as biomarkers of COPD. However, the utility of these potential markers in detecting COPD has not been validated. Moreover, how *Streptococcus parasanguinis*, N-acetylglutamate and its analogue N-carbamoylglutamate contribute to COPD pathogenesis has to be determined in animal models.

The authors should cite a paper where *Streptococcus Parasanguinis* has been isolated in COPD patients. Genome Anatomy of *Streptococcus Parasanguinis* Strain C1A, Isolated From a Patient With Acute Exacerbation of Chronic Obstructive Pulmonary Disease, Reveals Unusual Genomic Features. *Genome Announc*, 3 (3) 2015 May 28

The utility of these potential markers (*Streptococcus parasanguinis*, N-acetylglutamate and its analogue N-carbamoylglutamate) in detecting COPD has not been validated with experiments. What are the specificity and sensitivity?

What is really important is to show how *Streptococcus parasanguinis*, N-acetylglutamate and its analogue N-carbamoylglutamate contribute to COPD pathogenesis has to be determined in animal models. Without this information, the manuscript is just another paper describing changes in some microbiome and metabolites.

Spearman's rho was used but this is based on rank rather than absolute values.

More detailed information is required for all figure legends.

More specific titles for the figure legends can help. For example, the title for Fig. 4, "Multi-omic analysis of COPD-associated faecal microbiome and metabolome" is not sufficiently informative.

Figure 6. The pathway information on metabolites and OUT is confusing and not easy to process. Please use bar graphs and other traditional ways to plot the important findings. This applies also to other figures.

Reviewer #2 (Systems biology, microbiota, COPD)(Remarks to the Author):

Review of Bowerman et al. Chronic obstructive pulmonary disease patients have linked gut microbiome and metabolome changes associated with disease

This manuscript describes a comparative, cross-sectional study of the gastrointestinal tract (GIT) microbiome and metabolome in healthy and COPD subjects. The authors found some significant differences in several bacterial genera between the two groups, namely Streptococci. They also note that several GIT bacteria genera were correlated with biomarkers of COPD severity such as elevated neutrophil counts and reduced lung function. Untargeted metabolomics identified a COPD metabolite signature comprising 46% lipid, 20% xenobiotic and 20% amino acid related metabolites. Correlation

of the microbiome with the metabolome identified a disease associated network connecting *Streptococcus parasanguinis_B* with several metabolites associated with COPD. The author conclude that their work identifies new potential therapeutic targets and biomarkers COPD. They also claim this is the first study to analyze GIT microbiome status in COPD subjects.

The microbiome and its role in respiratory diseases is an intense area of investigation. In respiratory diseases, there are at multiple dynamic microbiome communities to consider – those of the respiratory tract, oral cavity and the GIT. For COPD, much effort has been focused on the respiratory tract microbiota, particularly in the lower tract or lung. Several studies have reported on a distinct microbiota changes associated with the severity of COPD exacerbation events, as cited by these authors. Both the lung and GIT tract have been studied in the context of non-COPD respiratory disorders, such as asthma, where changes in microbiota communities in both body sites have been associated with the disease. The GIT microbiome has been less studied in COPD, and in this respect, the present study is a useful contribution to the field.

However, while the overall bioinformatics and statistical analyses follow standardized work-flows and appear competently done for the most part, the study has several limitations which limit the support of the authors' overall conclusions.

Major critique:

1. The overall study is associative but not causal. In other words, there are associations with microbiome and metabolite variables with COPD vs healthy. However, the study cannot discern if these differences play a direct role in disease etiology or are the secondary consequences of the disease state. Therefore, the conclusion about using these results to design new therapeutics or biomarkers is over-extending the findings of this study.
2. With respect to sampling, the statistical power of the study is limited by the low number of subjects (28 COPD patients and 29 healthy controls) and only a single sampling event. In contrast, lung microbiome studies have shown that the time of sampling is critical with respect to lung exacerbation and recovery events. This manuscript does not mention if the COPD subjects were sampled with respect to their individual disease status and severity.
3. The lack of multiple samples over-time means that individual variability in the microbiome was not assessed. Without a defined time series of samples, dynamic change in the microbiome cannot be assessed within individuals. As previous studies have shown, the most significant signal is from overall changes within individuals in microbiome and composition diversity over time and in relationship to disease events (i.e. exacerbation, recovery, post-therapy, etc.). Overall, the authors need to have section in the Discussion which clearly states the limitations of their study design and how future studies should be implemented to address these gaps.
4. No differences in the microbiota were observed in terms of drug treatment, exacerbation frequency or smoking status. These findings differ from those based on the lung microbiome. Antibiotics as well as inhaled steroids (because a large portion of the drug ends up in the gut), have been shown to impact the GIT microbiota (see Maier et al. doi:10.1038/nature25979). The authors should comment on this discrepancy.
5. The finding that the differences in the metagenomes were non-significant yet glycosides and SecY in *Streptococcus* were slightly enriched in COPD, needs to be further explained. How was the significance of the *Streptococcus* pathways determined?

6. The claimed 77% accuracy of the random forest classifier for COPD status with a Kappa = 0.53, is moderate, at best. Other classifier algorithms such as XGBoost should be run as comparators.

7. The two metabolites with reduced levels in COPD patients relative to health were co-factors of carbamoyl phosphate synthetase I, the first enzyme in the urea cycle; the native co-factor, N-acetylglutamate, and its structural analogue, N-carbamoylglutamate. The authors conclude in the Discussion, "Hence, increasing fibre intake in COPD may be a relevant." The connection is not clear from the findings of the metagenomic or metabolite analyses, thus needs to be clarified.

Reviewer #1 (Gut immunity, metabolite)(Remarks to the Author):

Comment (C)1: *The authors examined 16S rRNA-based microbiome plus metagenomics and metabolome changes in COPD patients. Both microbiome and metabolomics analyses in COPD are fairly well established in the field. A uniqueness of this study is to identify the increase of N-acetylglutamate in COPD patients. The authors emphasize that Streptococcus parasanguinis, N-acetylglutamate and its analogue N-carbamoylglutamate could be used as biomarkers of COPD. However, the utility of these potential markers in detecting COPD has not been validated. Moreover, how Streptococcus parasanguinis, N-acetylglutamate and its analogue N-carbamoylglutamate contribute to COPD pathogenesis has to be determined in animal models. The authors should cite a paper where Streptococcus Parasanguinis has been isolated in COPD patients. Genome Anatomy of Streptococcus Parasanguinis Strain CIA, Isolated From a Patient With Acute Exacerbation of Chronic Obstructive Pulmonary Disease, Reveals Unusual Genomic Features. Genome Announc, 3 (3) 2015 May 28*

Response (R) 1: We thank the Reviewer for their time and consideration of our manuscript. We have addressed all of the comments below that has improved the manuscript. We disagree that that both microbiome and metabolomic analysis has been performed in the gastrointestinal tract in COPD patients. The lung microbiome/metabolome has been assessed but the gut factors have not^{1, 2, 3, 4}. Reviewer 2 agrees that this is useful contribution to the field (please see C7). We stated in the original submission:

Abstract:

“The lung microbiome is a contributing factor in COPD, however the role of the gut microbiome has not been defined.”

Introduction:

“However, the gastrointestinal microbiome of COPD patients has not been assessed.” – We have now added references 1 & 3 above as support for this statement (line 110).

This is the first comprehensive analysis of the gut microbiome in COPD, and the involvement of the gut-lung axis is a new concept in COPD that we are pioneering (Budden *Nat Rev Microbiol* 2017). It has the strong potential for therapeutic targeting that would be a whole new way to treat COPD. The Editor requested that we either provide additional patient or empirical data to validate the significance for COPD. We elected to collect and analyse samples from a validation cohort of patients (16 COPD patients, 22 healthy controls). This has validated and substantially improved the strength of our original findings. It would be interesting to determine how the bacteria and metabolites identified contribute to COPD pathogenesis in mouse models in future studies. This has now been highlighted in a new section on limitations in the Discussion (line 562-571):

“ Recognised variation in gut microbiome profiles between individuals and confounders such as medication status likely limited our ability to detect additional significant taxonomic and functional biomarkers for COPD. However, encouragingly there was a significant overlap between our relatively small study and validation cohorts. Analysis of larger COPD cohorts will likely identify additional significant correlated biomarkers. Our study was also limited to steady state disease and therefore did not capture the gut environment during disease exacerbation. Longitudinal analysis during exacerbation and recovery would be particularly interesting if paired with similar sampling of the lung environment to evaluate potential seeding from the gut. A design incorporating such repeated sampling of the same individual would also help overcome the problem of inter-individual variation.

Despite these limitations, a discriminatory signal is present in both the metagenomic and metabolomic datasets supporting the gut as a potential source of disease biomarkers in COPD. These candidates should be further evaluated for their mechanistic and causal involvement in COPD using established animal models^{5, 6, 7}.”

We have added reference to this genome announcement and reference, which is another lung microbiome study, as ref 59 (line 474).

C2: *The utility of these potential markers (Streptococcus parasanguinis, N-acetylglutamate and its analogue N-carbamoylglutamate) in detecting COPD has not been validated with experiments. What are the specificity and sensitivity?*

R2: The goal of this study was to define the capacity of gut-derived samples to distinguish COPD patients from healthy individuals, particularly the gut microbiome and metabolome. This is novel and there are no previous studies of the gut microbiome in the context of COPD. Our findings identify indicative taxa and metabolites

and correlate them with each other and disease features. Further work is needed to define their roles in disease, including characteristics such as specificity and sensitivity, however this work is beyond the scope of the current study. Please also see **R1**.

C3: *What is really important is to show how Streptococcus parasanguinis, N-acetylglutamate and its analogue N-carbamoylglutamate contribute to COPD pathogenesis has to be determined in animal models. Without this information, the manuscript is just another paper describing changes in some microbiome and metabolites.*

R3: We agree with the Reviewer that our findings require further study. However, this comment does not reflect the importance of our study which is the first in-depth study of the gut microbiome and its linked metagenome in COPD. In addition, we have now undertaken metagenomic analysis of a validation cohort comprising 38 individuals, the results of which are outlined in the last paragraph of the results section. The analysis finds overlap in COPD-associated species, with further similar trends observed. These results add substantial support for our original findings. While animal models are still required, these are beyond the scope of the present study. Please also see **R1**.

C4: *Spearman's rho was used but this is based on rank rather than absolute values.*

R4: Could the reviewer please provide some additional detail regarding their concern here? Spearman's rho was calculated using normalised mapping-based read counts per genome using the 'corr.test' function from the 'psych' R package. The function computes rank values from the given data using the base R function 'cor' and returns a correlation matrix and associated p-values.

C5: *More detailed information is required for all figure legends & More specific titles for the figure legends can help. For example, the title for Fig. 4, "Multi-omic analysis of COPD-associated faecal microbiome and metabolome" is not sufficiently informative.*

R5: We have modified the figure legends throughout, making the titles more informative.

C6: *Figure 6. The pathway information on metabolites and OUT is confusing and not easy to process. Please use bar graphs and other traditional ways to plot the important findings. This applies also to other figures.*

R6: Figure 6 displays correlation networks identified by mixOmics rather than pathway information. The connecting lines indicate correlation between two nodes, with the colour indicating the direction, either positive or negative. These graphs are typical of network maps, which necessarily have this structure, and are becoming more common in analyses of large datasets. However, to assist with interpretation, we have added bar graphs of the significantly differential nodes following covariate adjustment and improved the figure legend as per the above comment. The remaining figures contain horizontal bar graphs, or loadings plots, which represent the importance of different species or metabolites along a particular component of the sPLS-DA plots, with the longer bars representing more important features. We consider these plots are the best way of presenting this information, particularly as it quickly summarises a large number of features.

Reviewer #2 (Systems biology, microbiota, COPD)(Remarks to the Author):

C7: *Review of Bowerman et al. Chronic obstructive pulmonary disease patients have linked gut microbiome and metabolome changes associated with disease*

This manuscript describes a comparative, cross-sectional study of the gastrointestinal tract (GIT) microbiome and metabolome in healthy and COPD subjects. The authors found some significant differences in several bacterial genera between the two groups, namely Streptococci. They also note that several GIT bacteria genera were correlated with biomarkers of COPD severity such as elevated neutrophil counts and reduced lung function. Untargeted metabolomics identified a COPD metabolite signature comprising 46% lipid, 20% xenobiotic and 20% amino acid related metabolites. Correlation of the microbiome with the metabolome identified a disease associated network connecting Streptococcus parasanguinis_B with several metabolites associated with COPD. The author conclude that their work identifies new potential therapeutic targets and biomarkers COPD. They also claim this is the first study to analyze GIT microbiome status in COPD subjects. The microbiome and its role in respiratory diseases is an intense area of investigation. In respiratory diseases,

there are at multiple dynamic microbiome communities to consider – those of the respiratory tract, oral cavity and the GIT. For COPD, much effort has been focused on the respiratory tract microbiota, particularly in the lower tract or lung. Several studies have reported on a distinct microbiota changes associated with the severity of COPD exacerbation events, as cited by these authors. Both the lung and GIT tract have been studied in the context of non-COPD respiratory disorders, such as asthma, where changes in microbiota communities in both body sites have been associated with the disease. The GIT microbiome has been less studied in COPD, and in this respect, the present study is a useful contribution to the field.

However, while the overall bioinformatics and statistical analyses follow standardized work-flows and appear competently done for the most part, the study has several limitations which limit the support of the authors' overall conclusions.

R7: We thank the Reviewer for their time and consideration of our manuscript. We appreciate their recognition of the utility of our study. We have addressed the Reviewers comments as outlined below.

Major critique:

C8: 1. The overall study is associative but not causal. In other words, there are associations with microbiome and metabolite variables with COPD vs healthy. However, the study cannot discern if these differences play a direct role in disease etiology or are the secondary consequences of the disease state. Therefore, the conclusion about using these results to design new therapeutics or biomarkers is over-extending the findings of this study.

R8: We agree with this assessment by the Reviewer and have moderated our conclusions (lines 47, 110, 405, 440, 455, 562-575). Please also see **R1**.

C9: 2. With respect to sampling, the statistical power of the study is limited by the low number of subjects (28 COPD patients and 29 healthy controls) and only a single sampling event. In contrast, lung microbiome studies have shown that the time of sampling is critical with respect to lung exacerbation and recovery events. This manuscript does not mention if the COPD subjects were sampled with respect to their individual disease status and severity.

R9: The Reviewer raises an important consideration with respect to a disease characterised by exacerbation episodes. All samples in the current study from COPD patients were taken during periods of stable disease and this has been added (line 119). Further we have now added the GOLD stage classification for COPD subjects (Table S1 and associated text). We now also include a validation of cohort of 16 COPD patients and 22 healthy controls, increasing the power to 44 COPD patients and 51 healthy controls. This has consolidated the findings. We have also added statements to the Discussion addressing the fact that this is a cross-sectional study and that longitudinal studies will be valuable. Please see **R1**.

C10: 3. The lack of multiple samples over-time means that individual variability in the microbiome was not assessed. Without a defined time series of samples, dynamic change in the microbiome cannot be assessed within individuals. As previous studies have shown, the most significant signal is from overall changes within individuals in microbiome and composition diversity over time and in relationship to disease events (i.e. exacerbation, recovery, post-therapy, etc.). Overall, the authors need to have section in the Discussion which clearly states the limitations of their study design and how future studies should be implemented to address these gaps.

R10: Discussion of the limitations of the study have been added to the manuscript (lines 562-575). Please also see **R1** and **R9**.

C11: 4. No differences in the microbiota were observed in terms of drug treatment, exacerbation frequency or smoking status. These findings differ from those based on the lung microbiome. Antibiotics as well as inhaled steroids (because a large portion of the drug ends up in the gut), have been shown to impact the GIT microbiota (see Maier et al. doi:10.1038/nature25979). The authors should comment on this discrepancy.

R11: This is a valuable comment thank you. We consider the discrepancy here is likely due to a combination of sample size and the number of medications taken by the COPD group. 70% of the COPD patients were taking inhaled corticosteroids, making the negative steroid group small (n = 9). These nine also differed in their other

medications, eg proton pump inhibitors that are also likely to affect the gut microbiome, introducing too much variation to produce a discriminatory signal. We have added comment to this effect in the added 'limitations' paragraph in the Discussion (lines 562-575).

C12: 5. *The finding that the differences in the metagenomes were non-significant yet glycosides and SecY in Streptococcus were slightly enriched in COPD, needs to be further explained. How was the significance of the Streptococcus pathways determined?*

R12: We initially compared the global signal derived from read annotation with KEGG, CAZy, Pfam and Tigrfam databases using all annotations for the two groups using PERMANOVA, which could not distinguish between the groups. Following this, we undertook pairwise comparisons of each individual KEGG, Pfam etc annotation using DESeq2, which produced significant results for a subset of annotations. Reads annotated with these domains were then extracted and aligned to the genome database to determine their likely species of origin – mainly *Streptococcus*. We have now clarified the approach (lines 220-222). Following this gene-centric analysis, we also undertook a genome-centric comparative analysis of genomes identified as significantly enriched or depleted in COPD. This analysis compared the predicted functional potential of 60 genomes - 35 enriched vs 25 depleted – to examine trends amongst genomes either enriched or depleted in COPD. These analyses are complementary, with the gene-centric approach assessing the broader metagenome and the genome-centric focused on species associated with the condition of interest. We have now clarified that the genome-centric is distinct from the gene-centric approach (lines 257-259).

C13: 6. *The claimed 77% accuracy of the random forest classifier for COPD status with a Kappa = 0.53, is moderate, at best. Other classifier algorithms such as XGBoost should be run as comparators.*

R13: We attempted to improve the classification support level by optimising parameters of the random forest model however this was unsuccessful (max accuracy 81%, kappa 0.62). Furthermore, the model performed poorly when used to classify the validation cohort (accuracy 57%, kappa 0.23). We have therefore removed reference to this analysis from the manuscript.

C14: 7. *The two metabolites with reduced levels in COPD patients relative to health were co-factors of carbamoyl phosphate synthetase I, the first enzyme in the urea cycle; the native co-factor, N-acetylglutamate, and its structural analogue, N-carbamoylglutamate. The authors conclude in the Discussion, "Hence, increasing fibre intake in COPD may be a relevant." The connection is not clear from the findings of the metagenomic or metabolite analyses, thus needs to be clarified.*

R14: The suggestion regarding fibre intake derives from the dietary survey that showed a reduced fibre intake in COPD patients, summarised in Table S1a, rather than the metagenomic or metabolomic analyses. The source of the data has now been included (line 541).

Other changes we have made to improve the manuscript

Analysis supporting Figure 3 was updated to use centered log ratio transformed relative abundance values for consistency with analysis supporting Figure 5. This resulted in additional species being correlated with lung measurements with edits added to the manuscript (lines 42-43 and 206-211).

1. Chunxi L, Haiyue L, Yanxia L, Jianbing P, Jin S. The gut microbiota and respiratory diseases: New evidence. *J Immunol Res* **2020**, 2340670-2340670 (2020).
2. Nambiar S, Bong How S, Gummer J, Trengove R, Moodley Y. Metabolomics in chronic lung diseases. *Respirology* **25**, 139-148 (2020).
3. Vaughan A, Frazer ZA, Hansbro PM, Yang IA. COPD and the gut-lung axis: the therapeutic potential of fibre. *J Thorac Dis* **11**, S2173-S2180 (2019).
4. Ran N, *et al.* An updated overview of metabolomic profile changes in chronic obstructive pulmonary disease. *Metabolites* **9**, 111 (2019).
5. Beckett EL, *et al.* A new short-term mouse model of chronic obstructive pulmonary disease identifies a role for mast cell tryptase in pathogenesis. *J Allergy Clin Immunol* **131**, 752-762 (2013).

6. Hansbro PM, *et al.* Importance of mast cell Prss31/transmembrane tryptase/tryptase-gamma in lung function and experimental chronic obstructive pulmonary disease and colitis. *J Biol Chem* **289**, 18214-18227 (2014).
7. Jones B, *et al.* Animal models of COPD: what do they tell us? *Respirology* **22**, 21-32 (2017).

REVIEWERS' COMMENTS

Reviewer #1 (Remarks to the Author):

The authors revised the manuscript. The revised manuscript shows bacterial species and metabolites that are associated with COPD.

The manuscript is still highly descriptive, showing COPD-associated bacterial species and certain metabolites but no information is provided regarding how the levels of metabolites are altered in COPD and how this may regulate COPD pathogenesis.

In Figure 1, they are showing 16S rRNA analysis. Labeling is missing for X-axis Fig 1c.

In Figure 2, Metagenomic sequencing data are shown. In Panel b, labeling for X-axis is missing. Also, there is a grammar issue with the legend title:

"Fig. 2 Metagenomic sequencing-based exploration of COPD-associated faecal microbiota shows distinction from healthy individuals."

In most figures, the authors should show error bars showing the variations within COPD and healthy controls. This applies F1c, F2b, F3b, F3d, F7b and some of the supplementary figures.

Reviewer #2 (Remarks to the Author):

The authors have done an admirable job of responding to my initial comments on their manuscript (Reviewer 2). I have no other suggested revisions and support its publication in Nature Communications.

Thank you

REVIEWERS' COMMENTS

Reviewer #1 (Remarks to the Author):

The authors revised the manuscript. The revised manuscript shows bacterial species and metabolites that are associated with COPD.

The manuscript is still highly descriptive, showing COPD-associated bacterial species and certain metabolites but no information is provided regarding how the levels of metabolites are altered in COPD and how this may regulate COPD pathogenesis.

Response: We were requested by the Editor to either perform validation in another human cohort of complete experimental mechanistic studies to how the changes in taxa and metabolites may regulate pathogenesis. For this manuscript we performed the former and have validated our findings in a second separate cohort. We are performing mechanistic studies also but these will form another manuscript in the future.

In Figure 1, they are showing 16S rRNA analysis. Labeling is missing for X-axis Fig 1c.

Response: Labelling of the X-axis has been added.

In Figure 2, Metagenomic sequencing data are shown. In Panel b, labeling for X-axis is missing. Also, there is a grammar issue with the legend title:

“Fig. 2 Metagenomic sequencing-based exploration of COPD-associated faecal microbiota shows distinction from healthy individuals.”

Response: Labelling of the X-axis has been added. The title to Figure 2 title has been updated to: Metagenomic sequencing-based exploration of COPD-associated (n=28) faecal microbiomes supports distinction from those of healthy individuals (n=29).

In most figures, the authors should show error bars showing the variations within COPD and healthy controls. This applies F1c, F2b, F3b, F3d, F7b and some of the supplementary figures.

Response: These plots are visual representations of a single loading vector value for each element derived from the sPLS-DA plots accompanying them in F1b, F2a, F4a, F4c, F7a. There are therefore no error bars associated with these plots.

Reviewer #2 (Remarks to the Author):

The authors have done an admirable job of responding to my initial comments on their manuscript (Reviewer 2). I have no other suggested revisions and support its publication in Nature Communications.

Response: We thank the Reviewer for the recognition of our effort and their support of our manuscript.